# A de novo matrix for macroscopic living materials from bacteria

Sara Molinari ®[1], Robert F. Tesoriero Jr. ®[1,2], Dong Li[3], Swetha Sridhar ®[1,2], Rong Cai[1], Jayashree Soman[1], Kathleen R. Ryan ®[4], Paul D. Ashby ®[3] & Caroline M. Ajo-Franklin ®[1] ✉

Engineered living materials (ELMs) embed living cells in a biopolymer matrix to create materials with tailored functions. While bottom-up assembly of macroscopic ELMs with a de novo matrix would offer the greatest control over material properties, we lack the ability to genetically encode a protein matrix that leads to collective self-organization. Here we report growth of ELMs from *Caulobacter crescentus* cells that display and secrete a self-interacting protein. This protein formed a de novo matrix and assembled cells into centimeter-scale ELMs. Discovery of design and assembly principles allowed us to tune the composition, mechanical properties, and catalytic function of these ELMs. This work provides genetic tools, design and assembly rules, and a platform for growing ELMs with control over both matrix and cellular structure and function.

Naturally occurring living biomaterials, such as bones or wood, grow bottom-up from a small number of progenitor cells into macroscale structures[1]. Engineered living materials (ELMs)[2–4] are inspired by naturally occurring living materials, but use synthetic biology to introduce tailored, non-natural functions. By incorporating engineered cells into a biopolymer matrix, these materials can function as living sensors[5], therapeutics[6], biomanufacturing platforms[7], electronics[8], energy converters[9], and structural materials[10]. While cells confer functionality to ELMs, the matrix assembles the material and controls the bulk material composition, structure, and function[11].

Since the matrix plays such a key role in generating material properties, one primary goal of the field is to create ELMs that both have a synthetic biomolecular matrix—that can control these properties—and grow autonomously into macroscopic structures. However, such bottom-up, de novo ELMs are considered well beyond the current state-of- art[11] because secreting recombinant biopolymers at concentrations that gelate is challenging[12] and because the assembly of micrometer-sized cells into centimeter-scale materials requires self-organization across length scales spanning four orders of magnitude. Engineering principles to achieve this assembly are unknown[11,12]. Therefore, most macroscopic ELMs have been produced by adopting a top-down approach (such as 3D printing) to incorporate living cells into an exogenous matrix[6,13,14] or by processing microscopic ELMs that grow a synthetic biomolecular matrix into macroscopic materials[15–19]. The few autonomously produced, macroscopic ELMs have been created by genetically modifying existing nanocellulose matrices[20] or genetically manipulating the mineralization of silica matrices. However, these two approaches to autonomously produced, macroscopic ELMs have afforded little genetic control over the mechanical properties, e.g., ~1.2–1.4-fold change in the storage modulus[20,21]. This tunability is much more limited than the tunability of naturally occurring materials, chemically synthesized materials, or macroscopic ELMs produced by processing[22,23].

We posit that new strategies for developing synthetic biomolecular matrices to self-assemble bacteria into macroscopic ELMs can be informed by prior work on surface-engineered bacteria and surface-modified colloidal particles. The surface of *Escherichia coli* has been engineered to display interacting proteins, such as leucine zippers[24] or antigen-nanobody pairs[25], via outer membrane proteins. Engineered strains that display interacting pairs will self-assemble into cell–cell aggregates that flocculate[24,25]; however, these aggregates are microscopic and must be processed to form larger materials[18]. In contrast,

[1]Department of Biosciences, Rice University, Houston, TX, USA. [2]Systems, Synthetic and Physical Biology PhD program, Rice University, Houston, TX, USA. [3]Molecular Foundry, Lawrence Berkeley National Laboratory, Berkeley, CA, USA. [4]Plant & Microbial Biology, University of California, Berkeley, Berkeley, CA, USA. ✉e-mail: cajo-franklin@rice.edu

micron-sized colloidal particles (typically polystyrene) that display DNA have been programmed to self-assemble into both microscopic[21] and macroscopic crystalline solids[26]. Over two decades of work on these systems has established central principles that underlie their self-assembly[27]. One of these central principles is that the interactions between particles must be mediated by high-density surface modifications, e.g., 1 DNA molecule per 27 nm[2,26]. Since the outer membrane proteins used for bacterial adhesins are displayed at ~5% of this density, i.e., 1 nanobody per 640 nm[2,28], we hypothesized that a matrix composed of self-interacting proteins displayed on bacteria at high density could lead to the formation of macroscopic solid materials.

We have previously engineered the surface layer (S-layer) of the oligotrophic bacterium *Caulobacter crescentus* for high-density peptide display[29] and biopolymer secretion[23]. The S-layer forms a 2D crystalline layer on the extracellular surface of *C. crescentus*, opening the possibility of displaying proteins at a density of up to 1 protein per 70 nm[2,30]. Leveraging this prior work, here we describe the autonomous formation of macroscopic living material from *C. crescentus* engineered to display a synthetic, self-interacting, protein matrix based on the S-layer scaffold. We demonstrate that the mechanical properties of this material can be genetically controlled over a factor of ~25x. We also describe unexpected findings indicating that the protein matrix plays a multifaceted role in the material formation and that material assembly occurs through a multi-step process mediated by the air–water interface.

## Results

### De novo engineering of a macroscopic bottom-up ELM

Leveraging existing genetic tools in *C. crescentus*[29,31], we sought to create bottom-up ELMs composed of cells that interact at high density through a surface-bound, de novo matrix. To minimize native cell–cell interactions, we started with a *C. crescentus* background that lacks the adhesive holdfast and therefore cannot form a biofilm[32,33]; we refer to this as the wild-type strain. Next, we designed a displayed bottom-up de novo (BUD) protein by replacing the native copy of the surface layer (S-layer) RsaA[30] (Fig. 1a−top) with a synthetic construct encoding four functional regions (Fig. 1a−bottom): (i) a surface-anchoring domain for high-density display, (ii) a biopolymer region to influence the material properties, (iii) tags for functionalization, and (iv) a domain for self-interaction and secretion. To anchor the BUD protein on the extracellular surface of *C. crescentus*, we used the first 250 residues of RsaA, known to bind the O-antigen lipopolysaccharide at a density of ~14,000 copies per µm[2, 34–36]. For the biopolymer region, we chose an elastin-like polypeptide (ELP) based on human tropoelastin that contains 60 repeats of the Val-Pro-Gly-X-Gly motif [30,37] (ELP$_{60}$). ELPs form elastic materials, are flexible, and self-associate in a concentration-dependent fashion. Thus, this region is designed to influence material properties, promote solution accessibility, and add non-covalent self-interactions. SpyTag[38] was used as a functionalization tag, as it covalently binds to fusion proteins containing SpyCatcher. We also introduced a FLAG tag as an epitope marker. Lastly, the C-terminal domain of the BUD protein, consisting of the last 336 residues of RsaA, was chosen to mediate protein secretion[29] and to self-associate[39]. We refer to this BUD protein-expressing strain of *C. crescentus* as the BUD-ELM strain. In this way, we designed a *C. crescentus* strain to display a high-density, surface-bound, elastin-based matrix across its entire surface (Fig. 1b).

To test whether the BUD-ELM strain autonomously forms macroscopic material, we grew it and the wild-type strain in liquid culture using standard media and growth conditions for *C. crescentus*. The

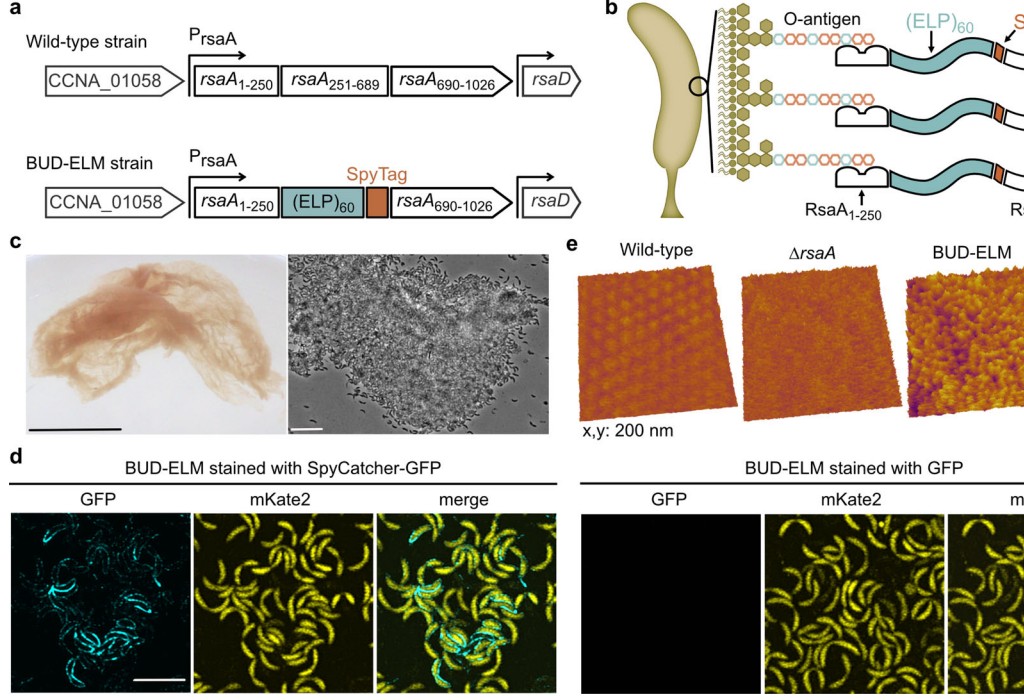

**Fig. 1 | Engineered strains of *C. crescentus* self-assemble into BUD-ELMs.**
**a** Schematic of the native *rsaA* gene within its genomic context, showing its N-terminal cell anchoring domain (*rsaA*$_{1−250}$) and C-terminal domain (*rsaA*$_{251−1026}$) with the secretion subdomain (*rsaA*$_{690−1026}$) (top). Schematic of the construct replacing the native *rsaA* gene in the BUD-ELM strain (bottom). **b** Illustration of the redesigned external surface of *C. crescentus*. showing the BUD protein attached to the cell surface. Absolute dimensions and relative positions of objects are only meant for illustrative purposes. **c** Photograph of free-floating material formed by

the BUD-ELM strain (left); the scale bar is 1 cm. Brightfield image of a portion of a BUD-ELM (right), showing cell clusters and intact cells; scale bar is 10 µm.
**d** Confocal microscopy of single cells of BUD-ELM strain stained with SpyCatcher-GFP (left) or GFP (right), demonstrating that the BUD protein is located on the cell surface. The scale bar is 5 µm and applies to every image. **e** AFM images of the cell surface of wild-type (left), Δ*rsaA* (middle), and BUD-ELM strain (right), showing the brush-like structure of the BUD-ELM strain's surface.

culture of wild-type *C. crescentus* lacking the BUD protein did not generate any visible aggregates after 24 h of growth (Supplementary Fig. 1). In contrast, cultures of the BUD-ELM strain yielded centimeter-scale, filamentous material (Fig. 1c−left) under identical conditions. The dry-mass yield of this material was $350 \pm 302$ mg material/L culture. The material contained intact *C. crescentus* cells (Fig. 1c−right) and cells aggregated into clumps with an area of greater than 50 μm². These aggregates are many times larger than spontaneous aggregates reported using other surface-display methods[24,25]. These data indicate our engineered *C. crescentus* strain autonomously grows into centimeter-scale ELMs.

To probe whether the BUD protein was surface-displayed and could play a role in BUD-ELM assembly, we compared the extracellular surface of planktonic cells of the BUD-ELM strain to its parental *C. crescentus* strains before material formation. When stained with Spy-Catcher-GFP, cells of the BUD-ELM strain (Fig. 1d−left) showed GFP fluorescence (cyan) along the outer contour of the cells (yellow), demonstrating the BUD protein is on the extracellular surface. The BUD-ELM was not stained by free GFP (Fig. 1d−right), nor was the Δ*SpyTag* strain (Supplementary Fig. 2a) stained by SpyCatcher-GFP (Supplementary Fig. 2b), confirming that staining required the specific binding between SpyTag and SpyCatcher. Atomic force microscopy (AFM) of the BUD-ELM cells showed a brush-like structure (Fig. 1e−right) that distinguished it from the wild-type hexameric S-layer (Fig. 1e−left) and the Δ*rsaA* strain (Fig. 1e−middle). The BUD protein formed long, unstructured projections, and this soft layer mediates cell−cell interactions (Supplementary Fig. 3). These results indicate that we encapsulated *C. crescentus* by a genetically-encoding high-density display of the BUD protein. They also provide a first demonstration of macroscopic, bottom-up ELMs with a de novo surface-bound matrix that mediates cell−cell interactions.

## BUD-ELMs are organized hierarchically through a synthetic proteinaceous matrix

To characterize the structure of BUD-ELMs across multiple length scales, we stained them with SpyCatcher-GFP and imaged them using confocal microscopy. At the half, a millimeter length scale (Fig. 2a−left), the BUD protein (cyan) and cells (yellow) appear distributed throughout the entire material. At the micron length scale, *C. crescentus* cells in the material display a layer of BUD protein (Fig. 2a−right), similar to planktonic cells of this strain (Fig. 1d). However, at the tens of micron length scale (Fig. 2a−middle), we unexpectedly observed a BUD protein-containing secreted matrix (blue) that was locally inhomogeneous and was surrounded by *C. crescentus* cells (yellow) on all sides (Supplementary Fig. 4). To probe the matrix composition, we also stained the BUD-ELM with Congo Red and 3,3'-dioctadecyloxacarbocyanine perchlorate (DiO) (Supplementary Fig. 5), which are known to bind amyloid proteins[40] and lipids[41], respectively. Congo Red staining (Supplementary Fig. 5−left) was orthogonal to the cell staining and overlapped with the SpyCatcher-GFP staining, confirming that the matrix is made of proteins. In contrast, DiO (Supplementary Fig. 5−right), did not stain cell-free regions. The absence of DiO staining excludes the hypothesis that the BUD-ELM matrix contains remains of lysed cells. Analysis of the cell-free and stained areas (Fig. 2b) confirmed that protein staining had a higher overlap with cell-excluded matrix regions compared to lipid staining. Thus, the BUD-ELM strain produces a secreted proteinaceous matrix containing the BUD protein that mediates the BUD-ELM structure at the tens of micron length scale.

To understand how the BUD protein ends up as both a surface-displayed and secreted matrix in the final BUD-ELM, we imaged single cells through AFM at the early stages of BUD-ELM formation, when cells are mostly in the planktonic state, and at later stages when the material is fully assembled. At the early stage (Fig. 2c−left), the cell surface appeared uniform, but after the BUD-ELM had formed, cells

showed large protuberances (Fig. 2c−right). Additionally, the surface layer depth of early-stage BUD-ELM cells is ~10 nm (Fig. 2d−left), compared to the ~35 nm layer of late-stage cells (Fig. 2d−right), indicating that the protein layer thickens over time. Hypothesizing that BUD proteins in this layer might be released from the cell surface as the layer thickens, we checked the extracellular medium of BUD-ELM cultures for the presence of the BUD protein by immunoblotting against the FLAG tag. We observed a band corresponding to the BUD protein at ~102 kDa, indicating that it is indeed present in the extracellular media. Note that wild-type RsaA migrates at a higher than expected apparent molecular weight (observed 113 vs. expected 98 kDa[30]), and the BUD protein shows this same apparent difference in molecular weight (observed 102 vs. expected 86 kDa). Interestingly, we found secreted BUD protein in the medium under both static and shaking conditions (Fig. 2e and Supplementary Fig. 6), indicating that some BUD protein is released into the medium independent of shaking. Nonetheless, these results demonstrate that the BUD protein is simultaneously present as a surface-displayed matrix protein and a secreted matrix protein during material formation.

Having established the roles of the BUD protein in BUD-ELM assembly, we next sought to understand the role of each domain of the BUD protein in assembly. To do so, we generated an additional strain lacking the $ELP_{60}$ (Δ$ELP_{60}$ BUD-ELM strain, Fig. 2f−left panel, top image) and compared it to the original BUD-ELM strain (Fig. 1a−bottom). The RsaA C-terminal domain could not be deleted because it is known to be essential for extracellular secretion. We observed that the Δ$ELP_{60}$ BUD-ELM strain (Fig. 2f −left panel, bottom-left image) forms BUD-ELMs that are very similar to the original BUD-ELM in morphology and color; optical microscopy also confirms that both are cell-rich materials (Fig. 2f−left panel, bottom images). Moreover, confocal microscopy shows that—despite the removal of the central $ELP_{60}$ domain−single cells are still surrounded by a layer of BUD protein (Supplementary Fig. 7). While the SpyCatcher-GFP staining is less intense (laser intensity was increased by 25% to visualize it), whole-cell immunoblotting indicates the amount of BUD protein attached to Δ$ELP_{60}$ cells is at least comparable to, if not greater, than the original BUD-ELM strain (Supplementary Fig. 8). This indicates the BUD protein lacking the $ELP_{60}$ region is less solvent-accessible than the original BUD protein. Taken together, these data show that the $ELP_{60}$ domain promotes solvent accessibility of BUD protein.

We also examined the role of the anchoring domain in material formation by using a previously described strain that lacks $RsaA_{1-250}$, but contains $ELP_{60}$ and $RsaA_{690-1026}$[29] (Fig. 2f−right panel, top image). This Δ$rsaA_{1-250}$ strain, where the BUD protein is only secreted but not displayed, formed centimeter-scale materials (Fig. 2f−right panel, bottom-left image). These materials are much lighter in color than the original strain, suggesting they contain fewer cells. Indeed, optical microscopy (Fig. 2f−right panel, bottom-right image) showed that this Δ$rsaA_{1-250}$ BUD-ELM has many fewer cells than the original BUD-ELM. Together, these data indicate that secreted BUD proteins critically enable centimeter-scale BUD-ELM assembly. Surface-displayed BUD proteins, on the other hand, allow the formation of cell-rich BUD-ELMs by facilitating cell−cell and cell-matrix interactions. Since we observe large cell−cell aggregates in the original BUD-ELM strain (Fig. 1c−left) that are spatially distinct from the secreted matrix (Fig. 2a), we suggest that this aggregation is promoted by high-density protein display. Thus, this work provides genetic design rules for both cell-rich and matrix-rich macroscopic engineered living materials that autonomously form.

## BUD-ELMs are formed through a multi-step process that depends on physical parameters

We next sought to understand how this material assembles by imaging BUD-ELM cultures at various times during their growth (Fig. 3a). Shaken cultures grew planktonically for ~12 h (Fig. 3a−left) before a thin

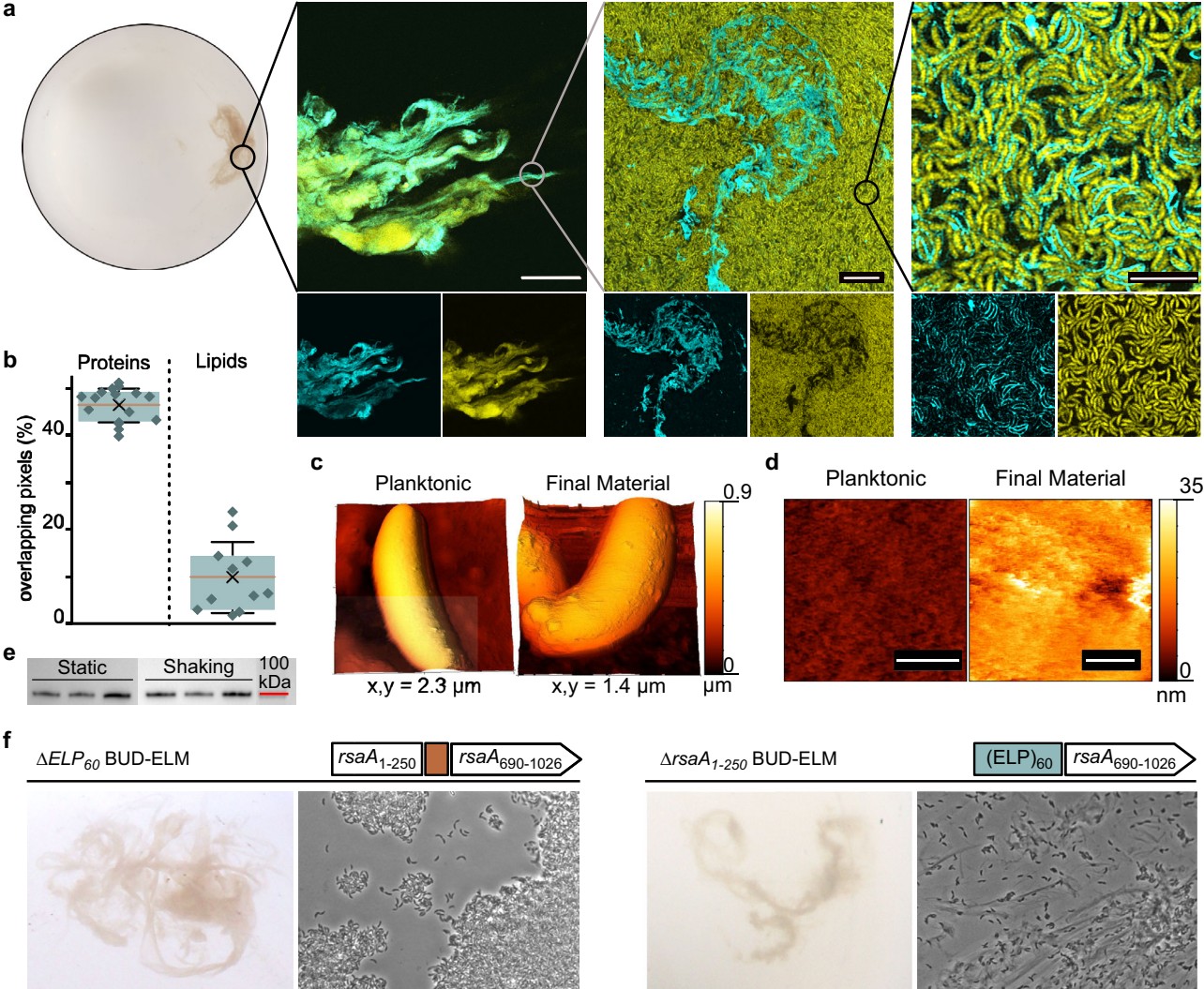

**Fig. 2 | BUD-ELMs contain a de novo protein matrix and display a hierarchical structure. a** Confocal microscopy of ELMs stained with SpyCatcher-GFP at increasing magnifications, showing a hierarchical structure. The bottom images show individual fluorescent channels: GFP (matrix) on the left and mKate2 (cells) on the right. Scale bars are, from left to right, 100, 10, and 5 μm, respectively. **b** Percentage of overlapping pixels between cell-free and stained regions, confirming the absence of lipids in the BUD protein matrix. Error bars are centered on the mean value (cross) and represent the standard deviation; the red line indicates the median value; the boxes show the interquartile range (25–75%). The analysis has been performed on 16 images for the protein and 11 for the lipid staining, each obtained from three independent samples. Source data are provided as a Source Data file. **c** AFM images of single cells at early (left) and late (right) stages of BUD-ELM formation, showing a difference in surface morphology. **d** High-resolution AFM images of single-cell surfaces at early (left) and late (right) stages of BUD-ELM formation, showing differences in surface layer thickness. Scale bars are 100 nm. **e** Immunoblot of BUD protein was detected in the growth media of culture grown in static (left) and shaking (right) conditions, showing a similar amount of protein in both conditions. BUD proteins were stained with the ANTI-FLAG® antibody. Source data are provided as a Source Data file. **f** Comparison between the $\Delta ELP_{60}$ and $\Delta rsaA_{1-250}$ BUD-ELM strains, showing differences in morphology and cell content. Each panel shows the genetic constructs (top), a representative image of BUD-ELMs at low (bottom, left), and high (bottom, right) magnification. For each panel, scale bars are 1 cm (bottom, left) and 50 μm (bottom, right).

pellicle appeared at the air–water interface (Fig. 3a–middle). AFM images of the pellicle depicted a central, cell-dense region (Fig. 3b–left) and a peripheral region of a few cells attached to a ~6 nm thick membrane (Fig. 3b–middle and right), suggesting the BUD protein forms a protein membrane to which cells adhere. The pellicle increased in density and opacity, becoming more compact. After ~24 h total culturing time, the pellicle desorbed from the air–water interface and sank as the final material (Fig. 3a–right). Disrupting the hydrophobicity of the air/water interface by the addition of surfactant prevented pellicle and material formation (Supplementary Fig. 9). Similarly, neither a pellicle nor material formed under static growth conditions. However, when static cultures were shaken, a pellicle formed (Supplementary Fig. 10). Together, these experiments demonstrate that BUD-ELMs are formed through a multi-step process

and establish hydrophobicity of the air/water interface and shaking as critical conditions for assembly of BUD-ELMs.

To understand and ultimately predict how physical parameters affect BUD-ELM assembly, we grew cultures in 125 and 250 mL flasks under different conditions and measured the size of the resulting materials. In order to do so, we imaged the bottom of the flasks where BUD-ELMs grew and quantified the flat surface area of the material; we defined this value as the apparent BUD-ELM size. The size of BUD-ELMs depended non-monotonically on the shaking speed, volume, and flask diameter (Fig. 3c). To develop a phenomenological model to describe this behavior, we used these parameters to calculate two quantities: the volumetric power input, $P_V$, describing the energy provided to the flask by shaking per unit volume and the volumetric mass transfer coefficient, $k_{La}$, representing the transfer of oxygen into the medium

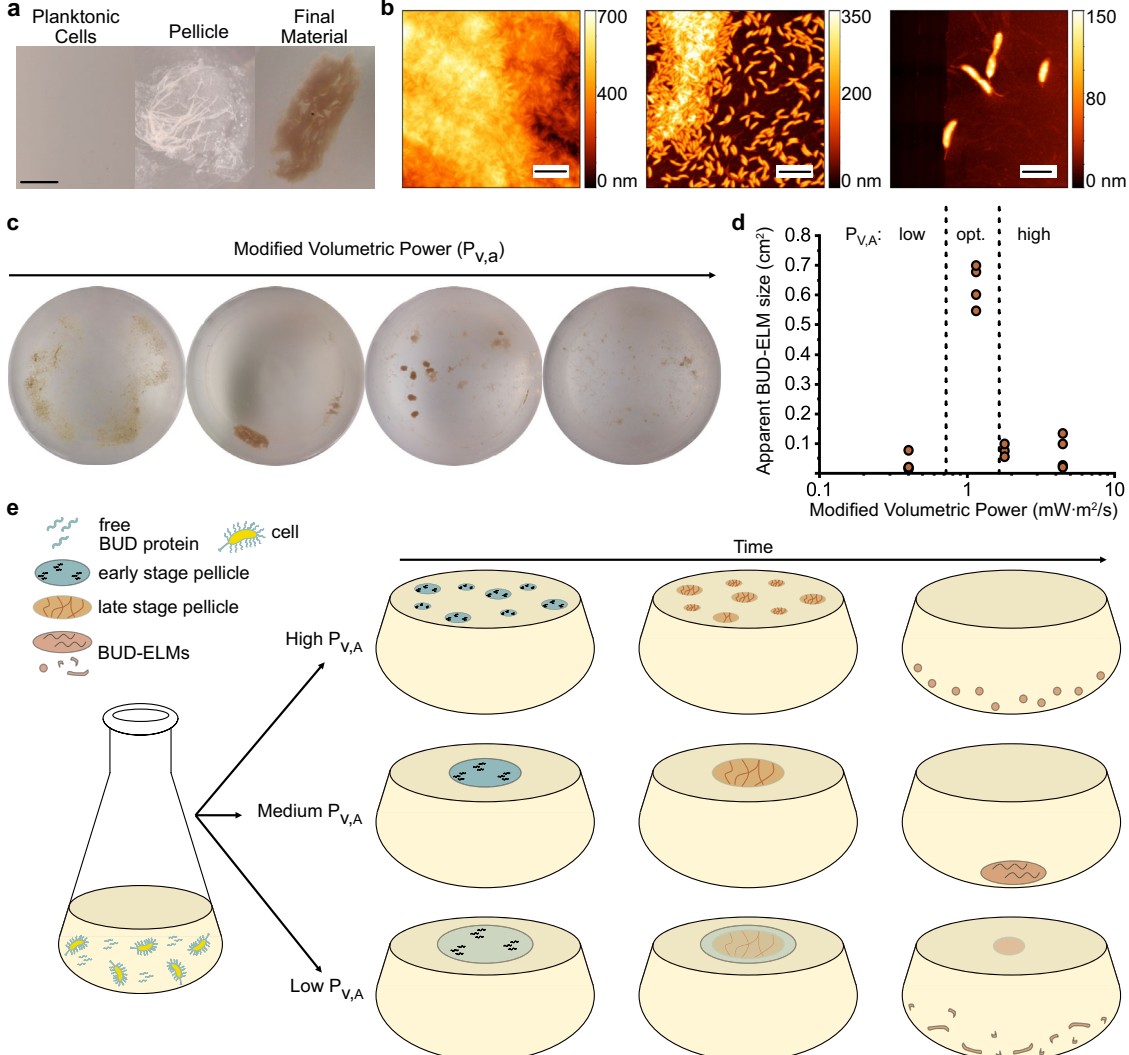

**Fig. 3 | BUD-ELMs are formed through a shaking-dependent, multi-step process. a** Optical images of representative BUD-ELM strain culture during material formation, showing BUD-ELMs are formed through a multi-step process. The scale bar is 1 cm. **b** AFM images of pellicle structure, showing the pellicle contains both a central region containing several layers of densely packed cells (left) and a peripheral region containing sparse cells connected by a thin membrane (center and right). Scale bars are, from left to right, 5, 5, and 2 μm, respectively. **c** Representative optical images of BUD-ELMs grown in 250 mL flasks under different modified volumetric power values. Altering the modified volumetric power changes the morphology and size of BUD-ELMs. **d** Correlation between modified volumetric power and the apparent surface area of BUD-ELMs grown in 500 mL shake flasks. Dotted lines separate the three ranges of $P_{V,A}$: low, optimal (intermediate) and high. The graph shows that the apparent surface of BUD-ELMs grown in a 500 mL area is small for low and high $P_{V,A}$ and larger at intermediate values of $P_{V,A}$, as predicted by the model (Supplementary Fig. 11b). Each $P_{V,A}$ condition was tested using at least three independent replicates. Source data are provided as a Source Data file. **e** Proposed mechanism for BUD-ELM formation, showing the effect of the modified volumetric power on the assembly of BUD-ELMs.

relative to the area of the air–water interface. We found that neither parameter showed a consistent relationship with the size of BUD-ELMs per flask (Supplementary Fig. 11a). Instead, we found empirically that the product of $P_V$, $k_{La}$, and the fifth power of the flask diameter, which we refer to as the modified volumetric power parameter, $P_{V,A}$, related the culture conditions to the material size (Supplementary Fig. 11b). Specifically, this model correlates the formation of the largest pieces of material with cultures grown within an optimal $P_{V,A}$ range of 0.72–1.65 mW m²/s.

To test whether this phenomenological model could accurately predict the size of material grown in larger flasks, we calculated shaking conditions for 500 mL flasks that would match $P_{V,A}$ values inside and outside the optimal range and used these conditions to grow BUD-ELM cultures in 500 mL flasks (Supplementary Table 4). In agreement with our model, cultures grown in the optimal $P_{V,A}$ range yielded material with large apparent sizes, while cultures outside this range yielded smaller material (Fig. 3d). This evidence indicates that

the optimal $P_{V,A}$ range can be applied to large-sized material production across different culture sizes and demonstrates our model can be used to scale-up cultures for BUD-ELM production.

With these data, we propose a model for BUD-ELM assembly (Fig. 3e). During culturing, the BUD protein accumulates in solution and on the surface of *C. crescentus*. With shaking, the BUD protein adsorbs to the air–water interface to form a protein-rich membrane of increasing thickness. BUD protein-displaying cells adhere to the membrane, increasing its density to form a pellicle. Hydrodynamic forces from shaking cause the pellicle to collapse on itself, until the material sinks to the bottom of the flask. At lower $P_{V,A}$ values, the weaker hydrodynamic forces lead to a thin pellicle that does not collapse, but instead gradually fragments into smaller materials. At intermediate $P_{V,A}$ values, stronger hydrodynamic forces collapse the pellicle into a single, large BUD-ELM. At higher $P_{V,A}$ values, shear forces prevent the assembly of larger pellicles, leading to independent smaller pellicles and smaller final materials. This empirical model

provides a basis for the future development of mechanistic models describing BUD-ELM assembly.

## BUD-ELMs are self-regenerating, multifunctional materials whose mechanical properties can be tuned genetically

Since the matrix plays an important role in determining the mechanics of biomaterials, and the matrix of BUD-ELMs is mostly composed of BUD proteins, we hypothesized that genetic manipulations of the BUD protein will have a significant effect on the mechanical properties of BUD-ELMs. To verify this hypothesis, we compared the mechanical properties of the original, $\Delta ELP_{60}$, and $\Delta rsaA_{1-250}$ BUD-ELMs. Rheological measurements (Supplementary Fig. 12) confirm that all three BUD-ELMs are viscoelastic solids. Frequency sweep curves (Supplementary Fig. 13) show large and significant differences in the storage modulus (G′) and the loss modulus (G″) among the three BUD-ELMs throughout the tested range of angular frequency. For a central value of angular frequency of 10 rad/s (Fig. 4a), the storage modulus of $\Delta ELP_{60}$ BUD-ELMs is increased by 4.4-fold of the original BUD-ELM, whereas the loss modulus increased by 4.0-fold. Conversely, the $\Delta rsaA_{1-250}$ BUD-ELMs show a 3.2-fold and 6.3-fold lower G′ and G″, respectively, relative to the original BUD-ELM. Comparing the $\Delta ELP_{60}$ BUD-ELMs to the $\Delta rsaA_{1-250}$ BUD-ELMs, we observe that these genetic changes can modulate the storage modulus and the loss modulus over 14-fold and 25-fold, respectively. We speculate that the increased stiffness of the $\Delta ELP_{60}$ BUD-ELMs reflects the removal of a long elastic linker, the $ELP_{60}$, from the BUD protein forming this cellular material. On the other end, we suggest $\Delta rsaA_{1-250}$ BUD-ELMs are less stiff due to the lack of crosslinking among cells and between the matrix and the cells. Overall, these results demonstrate that we can control BUD-ELMs mechanical properties over a 25-fold range through genetic modification of the matrix-forming BUD protein.

ELMs must be able to be processed and stored without losing their ability to regrow. We dried BUD-ELMs (Fig. 4b− left and middle) and re-inoculated fragments of them into a fresh medium (Fig. 4b−right). BUD-ELM fragments dried for 7, 14, or 21 days regenerated to form additional BUD-ELMs (Fig. 4c). Whereas BUD-ELMs re-grew 100% of the time after 7 or 14 days of drying, BUD-ELMs desiccated for 21 days regenerated in 33% of cases. Additionally, BUD-ELMs collected from multiple cultures formed a cohesive paste (Fig. 4d−left) that was extrudable through syringes with different diameters (Fig. 4d−two middle panels). When mixed with glass powder, BUD-ELMs behaved as cementing agents, creating a firmer paste that hardened into a solid composite (Fig. 4d−right). These results indicate that BUD-ELMs can regenerate after drying, can be reshaped, and can be processed into composite materials.

Lastly, we probed the ability of BUD-ELMs to behave as functional materials. Self-regenerating materials that remove heavy metals from water could help address the growing prevalence of heavy metal contamination. Since many forms of bacterial biomass non-specifically absorb heavy metals[42], we hypothesized that the BUD-ELM could remove $Cd^{2+}$ from the solution. When $0.013 \pm 0.007$ g of $\Delta SpyTag$ BUD-ELM was incubated for 90 min with a $CdCl_2$ solution of six ppb−1 ppb above the Environmental Protection Agency (EPA) limit−$90 \pm 5\%$ of cadmium was removed (Fig. 4e). While this material is not designed to have a larger sorption capacity than wild-type *C. crescentus cells,* these data show that the BUD-ELM has potential to be a much more useful tool for heavy metal removal than a suspension of single cells by virtue of being a macroscopic, solid material.

Next, we functionalized the BUD-ELM matrix to allow it to perform biological catalysis. We fused the oxidoreductase PQQ-glucose dehydrogenase (GDH), which couples oxidation of glucose to the reduction of a soluble electron carrier[43], to SpyCatcher. Cell lysates containing over-expressed *apo* SpyCatcher-GDH or GDH were reconstituted by adding the cofactor PQQ (pyrroloquinoline quinone) to obtain the *holo* forms of the enzyme. After confirming the activity of *holo* GDH in

both cases (Supplementary Fig. 14a), we observed that only BUD-ELMs incubated with SpyCatcher-*holo*-GDH enzymatically reduced an electron carrier (Fig. 4f and Supplementary Fig. 14b). This demonstrates that BUD-ELM can be functionalized directly from complex mixtures to act as catalysts. Together, these results show that BUD-ELMs can serve as versatile functional materials.

## Discussion

In summary, we developed macroscopic living materials that autonomously grow from engineered bacteria and that can be genetically encoded to have a wide range of mechanical properties. Specifically, we show that the expression of a self-interacting protein−the BUD protein−enables macroscopic material formation (Fig. 1 and Supplementary Fig. 1). When displayed on the cell surface, the BUD protein mediates drives cell−cell aggregation; when secreted into the media, the BUD protein forms an extracellular matrix that binds these aggregates into a centimeter-scale structure (Fig. 2). Assembly of these ELMs starts with the growth of the engineered strain as a predominately planktonic culture, followed by the formation of a pellicle and its ultimate collapse into a final material (Fig. 3). Importantly, understanding of these design and assembly rules enabled us to alter the mechanical properties of these ELMs by ~25-fold and to imbue them with catalytic properties (Fig. 4).

Our work identifies design rules that lead to the autonomous formation of BUD-ELMs and suggests other design rules to be tested. We identify a secreted matrix as a design constraint for this class of centimeter-scale, autonomously forming BUD-ELMs. Our work also indicates that a surface-anchored protein matrix is necessary for these materials to be cell-rich. Our data also suggests that this surface-anchored protein matrix may need to be present at high-density for cell-rich materials. This suggestion is supported by previous literature that shows that *E. coli* with self-interacting proteins displayed at ~10% the density of our engineered *C. crescentus* strains lead to small cell−cell aggregates[25]. However, additional studies that systematically vary the surface density are needed to test this hypothesis. Another design rule that will be critical to understand and explore in future work is the nature and strength of the self-interactions in the BUD protein. We selected the $RsaA_{690-1026}$ and $ELP_{60}$ domains because prior reports demonstrate they can self-aggregate[39,44]. However, additional studies are needed to identify the nature of self-interactions and their strengths in the existing BUD protein and the range of self-interactions that permit the assembly of macroscopic materials.

This work also identified assembly principles for the autonomous formation of macroscopic materials. We have demonstrated that nucleation of a pellicle at the liquid-air interface and hydrodynamically-driven coalescence and collapse of the pellicle are required to form macroscopic ELMs. Since pellicle formation is also a key step in nanocellulose-based living materials[18], we suggest that the use of the air−water interface to locally concentrate and order hydrophobic biomolecules into a matrix may represent a general assembly principle for macroscopic ELMs. The genetic tools and *C. crescentus* platform developed here will permit systematic exploration of design and assembly rules for programming the growth of centimeter-scale structures using living cells as building blocks.

By creating BUD-ELMs with a de novo, modular protein matrix, this work greatly expands the ability to tailor macroscopic ELMs for specific applications. One of the key advantages of the *C. crescentus* BUD-ELM platform developed herein is the highly reproducible, autonomous formation of engineered living materials. Growing BUD-ELMs from an engineered strain of *C. crescentus* requires only control of the temperature, media composition, flask and culture volume, shaking speed, and shaking orbit. We envision this simplicity will enable the ready adoption of this platform by other researchers. The

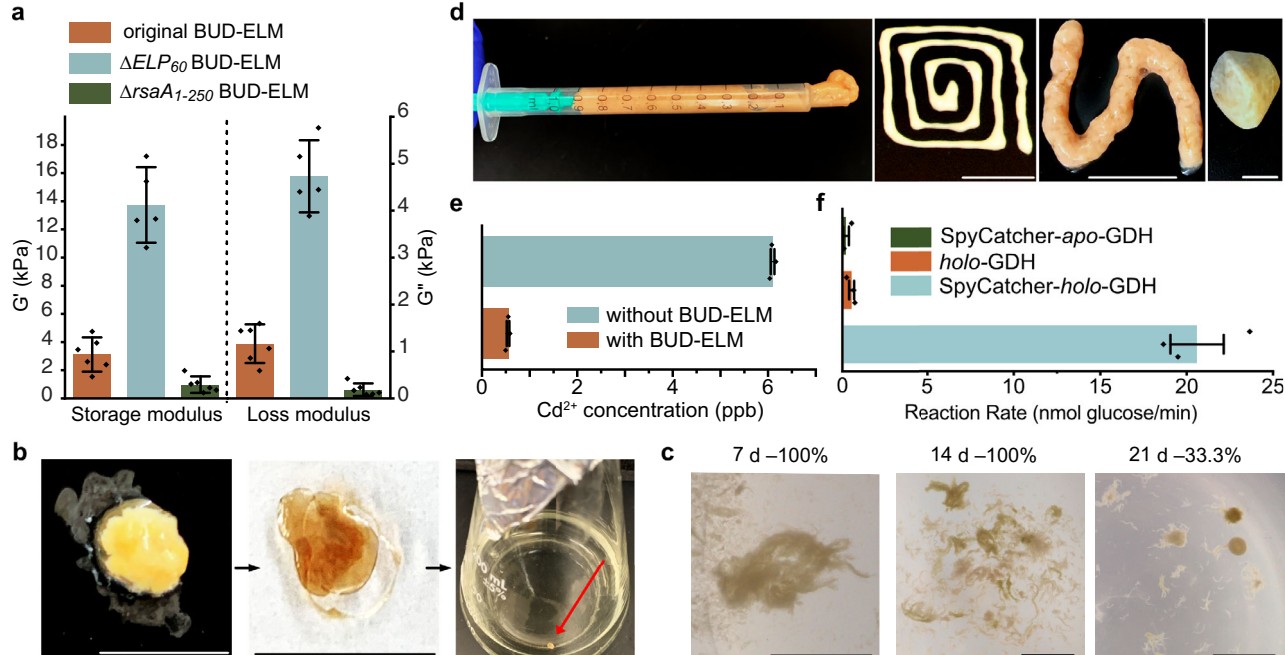

**Fig. 4 | BUD-ELMs are self-regenerating, processible, and functional materials.**
**a** Storage (G') and Loss (G'') modulus of original, $\Delta ELP_{60}$ and $\Delta rsaA_{1-250}$ BUD-ELMs
at an angular frequency of 10 rad s⁻¹, showing differences among the three BUD-
ELMs. Error bars are centered on the mean value and represent 95% confidence
intervals of at least five independent samples. Source data are provided as a Source
Data file. **b** Reseeding process of BUD-ELMs, showing extraction from liquid culture
(left), desiccated (middle) and inoculation into fresh medium (right). Scale bars are
1 cm. **c** Representative example of BUD-ELMs grown from desiccated material after
7 (left), 14 (middle), or 21 (right) days. The percentage of successful BUD-ELM
regeneration was 100, 100, and 33.3%, respectively. Percentages are calculated
from at least nine samples. Scale bars are 1 cm. **d** BUD-ELMs collected into a syringe
(left) for extrusion using different-sized nozzles (two middle panels), showing their
ability to be reshaped. Scale bars are 1 cm. BUD-ELMs are mixed with glass powder
to form a firm paste that hardens when dehydrated (right), showing its potential as
a cement-like agent. **e** Graph showing the final Cd²⁺ concentration after a six ppb
Cd²⁺ solution was incubated with or without $\Delta SpyTag$ BUD-ELMs. It shows that BUD-
ELMs are able to bind Cd²⁺ from aqueous solutions. Error bars are centered on the
mean value and represent the standard errors of three independent samples.
Source data are provided as a Source Data file. **f** Graph showing the rate of glucose
oxidation for BUD-ELMs that were incubated with SpyCatcher-*holo*-GDH, *holo*-
GDH, or SpyCatcher-*apo*-GDH. It confirms that BUD-ELMs specifically bind proteins
fused with SpyCatcher. Error bars are centered on the mean value and represent the
standard errors of three independent samples. Source data are provided as a
Source Data file.

second advantage of this platform is that the modularity of the BUD
protein and the ease of engineering protein biopolymers offer much
greater opportunities for introducing desirable properties into the
matrix[11]. The BUD-ELM variants described herein have storage moduli
that ranges between 13 kPa, comparable to nanocellulose-based
materials, and 0.5 kPa, comparable to printed curli fiber-based mate-
rials. Introducing sites for chemical crosslinking into the ELP domain
could allow the BUD-ELMs to be developed into elastomers[45]. More
broadly, this work enables leveraging known polypeptides and pro-
teins that exhibit desirable optical, electrical, mechanical, thermal,
transport, and catalytic properties[46]. We envision specific matrix
properties that can be combined synergistically with existing cellular
functions such as sensing, biomolecule production, and information
processing. Thus, this work multiplies the opportunities to program
ELMs tailored for applications in human health, energy, and the
environment.

## Methods
### Construction of BUD-ELM strains
All strains and plasmids used in this work are listed in Supplementary
Table 1 and 2, respectively. To generate all BUD-ELM strains (original–
RCC002, $\Delta ELP_{60}$–RCC004, and $\Delta SpyTag$–RCC005) from the wild-
type[47] (MFm126), we cloned integration plasmids designed to facilitate
the incorporation of synthetic DNA sequences into the *rsaA* locus
using homologous recombination. The integration plasmid used to
generate the original BUD-ELM strain (pSMCAF008) was cloned by
inserting a target sequence into the multicloning site of the backbone
plasmid pNPTS138 (GenBank: MK533795.1) using restriction enzymes

(ApaI upstream and NheI downstream). The target sequence (ordered
from GenScript, sequence below) encoded the $ELP_{60}$-SpyTag flanked
by 800 bp of homology regions up- and downstream of the native *rsaA*
central domain ($rsaA_{750-2073}$). The integration plasmids used to gen-
erate the $\Delta ELP_{60}$ and $\Delta SpyTag$ BUD-ELM strains (pSMCAF017 and
pSMCAF018, respectively) were cloned from the plasmid pSMCAF008
using Golden Gate assembly (PCR primers listed below).

The BUD-ELM strains were generated using the two-step
recombination technique[29]: plasmid pSMCAF008, pSMCAF017 or
pSMCAF018 was electroporated into *E. coli* WM3064 cells and subse-
quently conjugated overnight into *C. crescentus* NA1000 $\Delta sapA$::Pxyl-
mkate2 (MFm126) on a PYE agar plate containing 300n μM DAP. The
culture was then plated on PYE with 25 μg/ml kanamycin to select for
integration of the plasmid and removal of *E. coli* cells. Successful inte-
grants were incubated in liquid PYE media overnight and plated on PYE
supplemented with 3% w/v sucrose to select for excision of the plasmid
and *sacB* gene, leaving the target sequence in the genome. Integration
of the sequences was confirmed by colony PCR (with primers
SMCAF070 and SMCAF065) using a touchdown thermocycling proto-
col with an annealing temperature ranging from 72–62 °C, decreasing
1 °C per cycle. The PCR amplicons have been fully sequenced.

**Primers used to assemble pSMCAF017.** SMCAF142: TGGGTCTCAA
GGGCGGTTCGGGAGGAGGC

SMCAF143: TGGGTCTCAGCAATCCAAACGAGAGTCTAATAGAAT
GAGGTC

SMCAF144: TGGGTCTCCTTGCAACTGGTCTATTTTCCTCTTTTG

SMCAF145: TAGGTCTCCCCCCTTGTCATCGTCGTCCTTG.

**Primers used to assemble pSMCAFO18.** SMCAF168: CAGGTCTCTT GTCGCACCTGATTGCCCGA

SMCAF169: CAGGTCTCTGCTGGGACACCACCGCCAGG

SMCAF170: CTGGTCTCCCAGCTGACCCGGCCTTCGGC

SMCAF171: CTGGTCTCTGACAATCTATCGATTGTATGGGAAGC CCG.

**Primers used to amplify the modified *rsaA* gene in BUD-ELMs strains.** SMCAF070: TATGACGTTTGCTCTATAGCCATC

SMCAF065: GAGGATCAGACGTTCGCTTAG.

**GenScript target sequence.** CCAATGATCGTAATACGACTCACTAG TGGGGCCCGCGCCACTCGGTCGCAGGGGGTGTGGGATTTTTTTTGGG AGACAATCCTCATGGCCTATACGACGGCCCAGTTGGTGACTGCGTAC ACCAACGCCAACCTCGGCAAGGCGCCTGACGCCGCCACCACGCTGA CGCTCGACGCGTACGCGACTCAAACCCAGACGGGCGGCCTCTCGGA CGCCGCTGCGCTGACCAACACCCTGAAGCTGGTCAACAGCACGACG GCTGTTGCCATCCAGACCTACCAGTTCTTCACCGGCGTTGCCCCGTC GGCCGCTGGTCTGGACTTCCTGGTCGACTCGACCACCAACACCAACG ACCTGAACGACGCGTACTACTCGAAGTTCGCTCAGGAAAACCGCTTC ATCAACTTCTCGATCAACCTGGCCACGGGCGCCGGCGCCGGCGCGAC GGCTTTCGCCGCCGCCTACACGGGCGTTTCGTACGCCCAGACGGTC GCCACCGCCTATGACAAGATCATCGGCAACGCCGTCGCGACCGCCG CTGGCGTCGACGTCGCGGCCGCCGTGGCTTTCCTGAGCCGCCAGGC CAACATCGACTACCTGACCGCCTTCGTGCGCGCCAACACGCCGTTCA CGGCCGCTGCCGACATCGATCTGGCCGTCAAGGCCGCCCTGATCGG CACCATCCTGAACGCCGCCACGGTGTCGGGCATCGGTGGTTACGCGA CCGCCACGGCCGCGATGATCAACGACCTGTCGGACGGCGCCCTGTC GACCGACAACGCGGCTGGCGTGAACCTGTTCACCGCCTATCCGTCG TCGGGCGTGTCGGGTTCGGGCGGTTCGGGAGGAGGCTCGGGTGAC TACAAGGACGACGATGACAAGGGAGTTGGCGTCCCAGGAGTTGGAG TCCCAGGAGGGGGCGTTCCGGGCGCAGGAGTTCCTGGAGTAGGAGT TCCAGGAGTGGGCGTGCCAGGGGTGGGCGTCCCAGGTGGGGGAGTT CCCGGAGCAGGTGTGCCTGGGGGCGGCGTGCCTGGAGTCGGAGTTC CGGGGGTGGGTGTACCGGGTGGAGGCGTACCAGGCGCGGGAGTGC CGGGCGTGGGCGTGCCAGGCGTCGGTGTACCGGCGTTGGTGTTCC GGGCGGAGGTGTCCCCGGAGCTGGGGTTCCCGGTGGGGGTGTACCG GGCGTCGGGGTTCCCGGTGTGGGTGTCCCAGGTGGCGGCGTTCCCG GGGCGGGCGTACCTGGAGTGGGTGTGCCAGGAGTCGGCGTCCCAGG AGTCGGCGTACCAGGAGGTGGTGTTCCCGGGGCCGGAGTTCCCGGC GGAGGAGTTCCCGGCGTCGGCGTCCCTGGGGTCGGCGTCCCGGGAG GTGGAGTACCCGGAGCAGGAGTGCCGGGAGTCGGTGTACCTGGTGT CGGTGTCCCTGGTGTAGGTGTCCCGGGTGGTGGGGTGCCAGGTGCT GGCGTACCTGGGGGGGGGGTTCCTGGCGTAGGCGTTCCGGGGGTGG GCGTTCCGGGCGGCGGGGTGCCGGGAGCAGGTGTCCCCGGCGTTGG TGTACCGGGGGGTTGGTGTCCCAGGCGTAGGTGTGCCCGGTGGAGGG GTGCCGGGAGCTGGAGTGCCTGGAGGGGGTGTACCAGGGGTCGGTG TTCCCGGTGTAGGAGTACCGGGGGGCGGAGTCCCAGGAGCCGGCGT GCCGGGTGTTGGAGTCCCGGGAGTCGGAGTCCCTGGGGTAGGCGTT CCAGGGGGAGGGGTCCCCGGTGCAGGGGTTCCTGGCGGTGGTGTCC CAGGCGGTTCGGGAGGAGGCTCGGGTGCGCATATCGTAATGGTCGA TGCATACAAGCCCACGAAAGGAGGTTCAGGCGGCGGAAGCGGTGGT GGAAGCGGAGGTGGGTCAGGCGGAGGCTCAGGGGGAGGTTCGGGTG GCGGTTCGGGAGGAGGCTCGGGTGCTGACCCGGCCTTCGGCGGCTT CGAAACCCTCCGCGTCGCTGGCGCGGCGGCTCAAGGCTCGCACAAC GCCAACGGCTTCACGGCTCTGCAACTGGGCGCGACGGCGGGTGCGA CGACCTTCACCAACGTTGCGGTGAATGTCGGCCTGACCGTTCTGGCG GCTCCGACCGGTACGACGACCGTGACCCTGGCCAACGCCACGGGCA CCTCGGACGTGTTCAACCTGACCCTGTCGTCCTCGGCCGCTCTGGCC GCTGGTACGGTTGCGCTGGCTGGCGTCGAGACGGTGAACATCGCC GCCACCGACACCAACACGACCGCTCACGTCGACACGCTGACGCTGCA AGCCACCTCGGCCAAGTCGATCGTGGTGACGGGCAACGCCGGTCTGA ACCTGACCAACACCGGCAACACGGCTGTCACCAGCTTCGACGCCAGC GCCGTCACCGGCACGGGCTCGGCTGTGACCTTCGTGTCGGCCAACAC CACGGTGGGTGAAGTCGTCACGATCCGCGGCGGCGCTGGCGCCGAC TCGCTGACCGGTTCGGCCCACCGCCAATGACACCATCATCGGTGGCGC TGGCGCTGACACCCTGGTCTACACCGGCGGTACGGACACCTTCACGG GTGGCACGGGCGCGGGTATCTTCGATATCAACGCTATCGGCACCTCG ACCGCTTTCGTGACGATCACCGACGCCGCTGTCGGCGACAAGCTCGA CCTCGTCGGCATCTCGACGAACGGCGCTATCGCTGACGGCGCCTTCG GCGCTGCGGGTCACCCTGGGCGCTGCTGCGACGCTAGCTGACTGGGA AAACCCTGGCGTTAATCGGAAAGAACATGTGAGCAAAAGGCCAGCA AAAGGCCAGGAACCGTAAAAAGGCCGCGTTGCTGGCGTTTTTCCATA GGCTCCGCCCCCCTGACGAGCATCACAAAAATCGACGCTCAAGTCAG AGGTGGCGAAACCCGACAGGACTATAAAGATACCAGGCGTTTCCCCC TGGAAGCTCCCTCGTGCGCTCTCCTGTTCCGACCCTGCCGCTTACCG GATACCTGTCCGCCTTTCTCCCTTCGGGAAGCGTGGCGCTTTCTCAT AGCTCACGCTGTAGGTATCTCAGTTCGGTGTAGGTCGTTCGCTCCAA GCTGGGCTGTGTGCACGAACCCCCCGTTCAGCCCGACCGCTGCGCC TTATCCGGTAACTATCGTCTTGAGTCCAACCCGGTAAGACACGACTT ATCGCCACTGGCAGCAGCCACTGGTAACAGGATTAGCAGAGCGAGGT ATGTAGGCGGTGCTACAGAGTTCTTGAAGTGGTGGCCTAACTACGGC TACACTAGAAGAACAGTATTTGGTATCTGCGCTCTGCTGAAGCCAGT TACCTTCGGAAAAAGAGTTGGTAGCTCTTGATCCGGCAAACAAACCA CCGCTGGTAGCGGTGGTTTTTTTGTTTGCAAGCAGCAGATTACGCGC AGAAAAAAAGGATCTCAAGAAGATCCTTTGATCTTTTCTACGGGGTC TGACGCTCAGTGGAACGAAAACTCACGTTAAGGGATTTTGGTCATGA GATTATCAAAAAGGATCTTCACCTAGATCCTTTTAAATTAAAAATGAA GTTTTAAATCAATCTAAAGTATATATGAGTAAACTTGGTCTGACAGTT ACCAATGCTTAATCAGTGAGGCACCTATCTCAGCGATCTGTCTATTTC GTTCATCCATAGTTGCCTGACTCCCCGTCGTGTAGATAACTACGATAC GGGAGGGCTTACCATCTGGCCCCAGTGCTGCAATGATACCGCGAGAC CCACGCTCACCGGCTCCAGATTTATCAGCAATAAACCAGCCAGCCG GAAGGGCCGAGCGCAGAAGTGGTCCTGCAACTTTATCCGCCTCCATC CAGTCTATTAATTGTTGCCGGGAAGCTAGAGTAAGTAGTTCGCCAGT TAATAGTTTGCGCAACGTTGTTGCCATTGCTACAGGCATCGTGGTGT CACGCTCGTCGTTTGGTATGGCTTCATTCAGCTCCGGTTCCCAACGA TCAAGGCGAGTTACATGATCCCCCATGTTGTGCAAAAAAGCGGTTAG CTCCTTCGGTCCTCCGATCGTTGTCAGAAGTAAGTTGGCCGCAGTGT TATCACTCATGGTTATGGCAGCACTGCATAATTCTCTTACTGTCATGC CATCCGTAAGATGCTTTTCTGTGACTGGTGAGTACTCAACCAAGTCA TTCTGAGAATAGTGTATGCGGCGACCGAGTTGCTCTTGCCCGGCGTC AATACGGGATAATACCGCGCCACATAGCAGAACTTTAAAAGTGCTCA TCATTGGAAAACGTTCTTCGGGGCGAAAACTCTCAAGGATCTTACCG CTGTTGAGATCCAGTTCGATGTAACCCACTCGTGCACCCAACTGATC TTCAGCATCTTTTACTTTCACCAGCGTTTCTGGGTGAGCAAAAACAG GAAGGCAAAATGCCGCAAAAAAGGGAATAAGGGCGACACGGAAATGT TGAATACTCATACTCTTCCTTTTTCAATATTATTGAAGCATTTATCAG GGTTATTGTCTCATGAGCGGATACATATTTGAATGTATTTAGAAAAAT AAACAAATAGGGGTTCCGCGCACATTTCCCCGAAAAGTGCCACCTGA CGTC.

## Growth conditions of BUD-ELMs

Unless indicated otherwise, BUD-ELMs were grown by inoculating a single colony of *C. crescentus* strains into 80 mL of PYE in a 250 mL glass flask. All cultures were grown in an Innova 44 incubator shaker with a 2-inch orbit. These cultures were grown at 30 °C at 250 rpm, and BUD-ELMs typically formed within ~24–30 h. To explore the effect of growth parameters on BUD-ELM size, the flask volume, shaking speed, and culture volume were varied from 125 to 500 mL, 0 to 250 rpm, and 25 to 160 mL, respectively. The complete list of conditions tested can be found in Supplementary Tables 3 and 4.

## BUD-ELM desiccation and reseeding

To test the ability of BUD-ELMs to re-seed their own growth, BUD-ELMs grown under standard conditions were collected, transferred in a petri dish for 7, 14, or 21 days and left on the bench at room temperature. The material dried out in 24–48 h. To re-seed material growth, a ~0.3 to 0.5 cm² piece was broken off from the desiccated

material and inoculated into 80 mL of fresh PYE and grown under standard conditions. We detected BUD-ELM formation in 48 h for material dried over 7 or 14 days, and in 72 h for material dried over 21 days.

## Plasmid assembly for protein expression from *E. coli*

Plasmids pSMCAF015 and pSMCAF016 used for the expression of GFP and SpyCatcher-GFP from *E. coli* were assembled from existing constructs (pBAD-RFP and pBAD-SpyCatcher-RFP[29], respectively) by substituting the mRFP sequence with the GFP sequence (below). Similarly, plasmids pSMCAF032 and pSMCAF029 used for the expression of GDH and SpyCatcher-GSH from *E. coli* were assembled by introducing the GDH sequence in the same position. These plasmids were transformed into chemically competent BL21(DE3) cells (New England Biolabs – C2527H); single transformants were selected using ampicillin resistance.

**GFP sequence.** ATGCGTAAAGGCGAAGAGCTGTTCACTGGTGTCGTC CCTATTCTGGTGGAACTGGATGGTGATGTCAACGGTCATAAGTTTTC CGTGCGTGGCGAGGGTGAAGGTGACGCAACTAATGGTAAACTGAC GCTGAAGTTCATCTGTACTACTGGTAAACTGCCGGTTCCTTGGCCG ACTCTGGTAACGACGCTGACTTATGGTGTTCAGTGCTTTGCTCGT TATCCGGACCATATGAAGCAGCATGACTTCTTCAAGTCCGCCATGCC GGAAGGCTATGTGCAGGAACGCACGATTTCCTTTAAGGATGACGGCA CGTACAAAACGCGTGCGGAAGTGAAATTTGAAGGCGATACCCTGGTA AACCGCATTGAGCTGAAAGGCATTGACTTTAAAGAAGACGGCAATA TCCTGGGCCATAAGCTGGAATACAATTTTAACAGCCACAATGTTTA CATCACCGCCGATAAACAAAAAAATGGCATTAAAGCGAATTTTAAAA TTCGCCACAACGTGGAGGATGGCAGCGTGCAGCTGGCTGATCACTA CCAGCAAAACACTCCAATCGGTGATGGTCCTGTTCTGCTGCCAGAC AATCACTATCTGAGCACGCAAAGCGTTCTGTCTAAAGATCCGAACGA GAAACGCGATCATATGGTTCTGCTGGAGTTCGTAACCGCAGCGGGC ATCACGCATGGTATGGATGAACTGTACAAATAA.

**GDH sequence.** GACGTTCCGCTGACCCCGAGCCAGTTTGCGAAAGC GAAAAGCGAGAACTTCGACAAAAAAGTCATCCTGAGCAACCTGAATA AACCGCACGCTCTGCTGTGGGGTCCGGATAATCAGATTTGGCTGACC GAACGCGCAACCGGTAAAATTCTGCGCGTTAACCCGGAAAGCGGCA GCGTTAAAACCGTCTTTCAGGTTCCGGAAATCGTTAACGACGCAGA CGGTCAAAACGGTCTGCTGGGTTTTGCGTTTCATCCGGACTTCAAA AACAACCCGTACATCTACATCAGCGGCACCTTCAAAAAACCCGAAAAG TACCGACAAAGAGCTGCCGAATCAGACCATCATCCGTCGCTATACCT ACAACAAAAGCACCGACACCCTGGAAAAACCGGTTGATCTGCTGGCA GGTCTGCCGAGTAGTAAAGATCATCAGAGCGGTCGTCTGGTAATTG GTCCGGACCAGAAAATCTACTATACCATTGGCGATCAGGGCCGTAAC CAACTGGCATACCTGTTTCTGCCGAACCAAGCACAACATACCCCGAC CCAACAAGAACTGAACGGCAAAGACTACCACACCTACATGGGCAAA GTTCTGCGTCTGAATCTGGACGGTAGCATTCCGAAAGACAACCCGAG CTTCAACGGCGTTGTTAGCCATATCTATACCCTGGGTCACCGTAATC CGCAAGGTCTGGCATTTACCCCGAACGGTAAACTGCTGCAGTCTGAA CAGGGTCCGAATTCTGACGACGAAATCAACCTGATCGTTAAAGGCGG CAATTACGGTTGGCCGAACGTTGCAGGCTATAAAGACGATAGCGGCT ATGCATACGCGAATTATAGCGCAGCGGCAAACAAAAGCATCAAAGA CCTGGCCCAGAACGGTGTTAAAGTTGCAGCAGGCGTTCCGGTTACC AAAGAAAGCGAGTGGACCGGCAAAAACTTTGTTCCGCCGCTGAAAA CCCTGTATACCGTCCAGGACACCTACAACTATAACGATCCGACCTGC GGCGAAATGACCTATATTTGCTGGCCGACCGTTGCACCGAGTTCTG CATACGTTTACAAAGGCGGCAAAAAAGCGATCACCGGTTGGGAAAAT ACCCTGCTGGTTCCGAGTCTGAAACGCGGCGTTATCTTCCGCATCAA ACTGGATCCGACCTATAGTACCACCTACGACGATGCCGTTCCGATG TTCAAAAGCAACAACCGTTATCGCGACGTTATTGCAAGTCCGGACGG TAACGTTCTGTACGTTCTGACCGATACCGCCAGGTAACGTTCAGAAAG ACGACGGTAGCGTTACCAATACCCTGGAAAATCCGGGTAGCCTGAT CAAATTCACCTACAAAGCGAAATGA.

## Expression and purification of SpyCatcher-GFP, and GFP from *E. coli*

Single colonies of *E. coli* BL21(DE3) harboring plasmids pSMCAF015 and pSMCAF016, for expression of GFP and SpyCatcher-GFP, respectively, were inoculated in 25 mL of RM minimal media with 0.2% w/v glucose and 100 μg/mL ampicillin. After ~16 h of growth at 37 °C and 250 rpm, cells were used to inoculate 0.5 L of RM minimal media with 0.2% v/v glycerol, 100 μg/mL ampicillin and 0.0004% antifoam (Antifoam 204) to a final $OD_{600}$ ~0.05. The cultures were allowed to grow at 37 °C until mid-log phase. Protein production was induced with 0.2% w/v L-arabinose with incubation at 30 °C for ~17 h.

Cells were harvested by centrifugation at $8000 \times g$ for 30 min, resuspended in lysis buffer (50 mM Tris pH 8.0, 300 mM NaCl, 5% v/v glycerol, and 10 mM Imidazole) and lysed using Avestin Emulsiflex C3 Homogenizer. The lysate was centrifuged at $12,000 \times g$ for 1 h and the supernatant was collected for protein purification. The proteins were purified using Immobilized Metal Affinity Chromatography (IMAC) with a HisTrap FF column and buffers containing 50 mM Tris pH 8.0, 300 mM NaCl, 5% v/v glycerol, and 10–250 mM Imidazole. After protein purity was confirmed by SDS-PAGE, the protein was dialyzed into TEV-cleavage buffer (50 mM Tris pH 8.0, 0.5 mM EDTA, 1 mM DTT) and the 6 x His-tag was cleaved using TEV protease by agitation at 4 °C for 4 h. The cleaved protein was stored at −80 °C in 50 mM $NaPO_4$ pH 8.0, 300 mM NaCl, and 5% v/v glycerol.

## Determination of BUD-ELM dry weight

Eight BUD-ELMs were grown under standard conditions. Samples were harvested from liquid cultures, placed in Eppendorf tubes, washed once in $ddH_2O$ and lyophilized for 5 h in a Labconco Freezone 4.5 freeze dryer. Tubes were then weighted.

## In situ atomic force microscopy (AFM) imaging of single *C. crescentus* cells

Poly-L-lysine coated silicon substrates were immersed in Falcon™ round-bottom polypropylene culturing tubes containing 3 mL fresh *C. crescentus* cell culture at an $OD_{600}$ of 0.3–0.5. Culture tubes were then centrifuged at $3000 \times g$ for 10 min to immobilize the cells onto the silicon substrate. The silicon substrate was washed with 2 mL of sterile PYE to remove loosely-bound cells before being mounted to a metal puck and transferred to the AFM sample stage. In situ AFM was performed on an Asylum Cypher AFM using soft tapping mode. A fluid cell and two syringe pumps were assembled to control liquid flow and PYE medium was supplied to maintain cell viability during imaging. The AFM probe consisted of a sharp silicon tip on a silicon nitride cantilever (BioLever mini, BL-AC40TS) with a spring constant of 0.09 N/m. Cells were imaged in native state without fixation. A 100–200 mV amplitude setpoint was used to apply minimum forces (~0.2 nN) to cells during imaging.

## AFM imaging of pellicle structures

To bind the BUD-ELM pellicle to a silicon substrate, a 2 cm² precleaned silicon substrate was dipped into the pellicle forming cell culture with an entry angle of ~60° perpendicular to the water surface. The silicon substrate was then retrieved, and the pellicle structure was dried under an $N_2$ atmosphere for 2 h. The dried pellicle structure on the substrate was mounted to a metal puck and transferred to the AFM sample stage. AFM imaging was performed on an Asylum Cypher AFM using soft tapping mode in air. A Tap-150 tip (BudgetSensors) with a 5 N/m force constant was used to image the pellicle structure.

## Optical microscopy of BUD-ELMs

Small pieces of BUD-ELMs grown under standard conditions were placed between a slab of PYE agarose (1.5% w/v) and a glass coverslip-bottomed 50-mm Petri dish with a glass diameter of 30 mm (MatTek

Corporation) and imaged with an optical inverted microscope. Optical microscopy data were acquired using the software NIS-Elements AR (version 4.51.01). All microscopy pictures presented were generated using ImageJ software (version 2.0.0-rc-69/1.52p).

## Confocal microscopy of BUD-ELMs

Single colonies of BUD-ELM strain (RCC002) were inoculated in 30 mL PYE with 0.15% ᴅ-xylose—to induce the expression of mKate2, in a 125 mL flask and grown for 24 h at 30 °C at a shaking speed of 250 rpm. BUD-ELMs of similar dimensions were collected and washed twice with 1 mL of 0.01 M Phosphate-buffered saline (PBS), in a centrifuge tube. They were then incubated in 1 mL of 0.01 M PBS, at 30 °C, with the following staining agent: 80 µg of SpyCatcher-GFP or GFP for 1 h, 1% Congo Red (Thermo Fisher Scientific—D275) or 100 µg DiO (DiOC18(3) − 3,3′-Dioctadecyloxacarbocyanine Perchlorate) for 20 min. Samples were washed three times with 1 mL 0.01 M PBS and then a small amount was placed between a slab of PYE agarose (1.5% w/v) and a glass coverslip-bottomed 50-mm Petri dish with a glass diameter of 30 mm (MatTek Corporation). To acquire the low-magnification images (Fig. 2a−left panel), BUD-ELMs were embedded into 5% w/v agarose and sliced. The slice was placed on a glass coverslip-bottomed 50-mm Petri dish. For imaging, we used the Zeiss LSM800 Airyscan confocal microscope. Data were acquired using the ZEN (version 2.6) software and analyzed using ImageJ software (version 2.0.0-rc-69/1.52p). For Congo Red imaging, *C. crescentus* cells were transformed with a GFP-expressing plasmid (KR12) to distinguish the matrix from the cells.

## Immunoblot analysis of BUD proteins

For immunoblot analysis of culture supernatant, cultures of *C. crescentus* BUD-ELM strain (RCC002) were cultured in standard (shaking) or static (not shaking) conditions until they reached stationary phase (-OD 0.8 and 0.4, respectively). The supernatant of each culture was extracted and loaded onto a TGX Stain-Free™ gel (Biorad). After running, the gel was transferred to a 0.2 µm nitrocellulose membrane and blocked for 1 h at room temperature with SuperBlock™ blocking buffer (Thermo Scientific). Membranes were then washed four times in TBST buffer before incubation in a 1:5000 dilution of Monoclonal ANTI-FLAG® antibody (Monoclonal ANTI-FLAG® M2-Peroxidase (HRP) antibody from Sigma Aldrich − A8592-.2MG) produced in mouse, clone M2, purified immunoglobulin, buffered aqueous glycerol solution) solution for 1 h at room temperature. Membranes were washed an additional four times in TBST buffer before Clarity Max Western ECL Substrate (Biorad) was applied and the membrane was imaged for chemiluminescence. Each sample lane represents an independent sample, grown from separate individual colonies.

For whole-cell immunoblot analysis, the original and *ΔELP₆₀* BUD-ELM strains were grown under standard conditions. Planktonic cells from each colony were collected and washed 3 times in fresh PYE media to remove any free BUD protein not attached to the cell surface. After washes, all samples were normalized to an $OD_{600}$ value of 0.225 in 0.5 x PYE, 1 x Laemmli buffer. Each sample was then boiled for 5 min, and 10 uL was loaded onto a TGX Stain-Free™ gel (Biorad). All subsequent steps follow the protocol for immunoblotting described above. Protein molecular weight has been determined by image analysis with ImageJ.

## Apparent BUD-ELM size measure

To determine the apparent BUD-ELM size, the bottom of glass flasks where BUD-ELM grew was imaged using a Canon EOS 77D camera. Flasks were positioned within a reflective photobox on a clear plastic surface such that the bottom of the flask stood approximately 11.5 cm above the camera lens. These images were separated into RGB channels using MATLAB R2020b. A subset of

the blue channel images was then input into the image classification software ilastik (version 1.3.3), as a training set for the autocontext workflow. The first stage of training separated images into three different classifications: background, scattered material, and bundled material. Scattered material was defined as overlapping regions of small aggregates not associated with each other, whereas bundled material referred to larger, connected pieces of material. The second stage of training distinguished bundled material from the rest of the image. Both stages of training utilized all 37 features provided within the ilastik workflow (Ilastik, version 1.3.3). From the results of the training set, the second stage segmentation masks for all blue channel images were calculated, and loaded into MATLAB R202b. From these masks, the flat area of each piece of material was calculated, and the top five percentile of size from each image was averaged to yield a representative size measurement. For each image, a conversion rate between pixels and squared centimeters was determined using the standard flask diameter as a reference point. Size measurements were averaged between samples and plotted with respect to their calculated $(d5)(kLa)(P/V_L)$ values.

## Building of predictive parameter for BUD-ELM size

To describe the effect of shaking on BUD-ELM formation, a model was built based on the volumetric power input of a flask. The volumetric power input, defined as the rate of energy transfer into a flask per unit volume, was described by ref. 48 as:

$$\frac{P}{V_L} = \frac{Ne' \cdot \rho \cdot n^3 \cdot d^4}{V_L^{\frac{2}{3}}}, \tag{1}$$

Where $P$ is power, $V_L$ is culture volume, $n$ is shaking frequency, $d$ is the inner flask diameter, and $Ne'$ is the modified Newtons number. $Ne'$ was written in ref. 48 as a function of the Reynolds number $Re'$ in the following manner:

$$Re = \frac{\rho \cdot n \cdot d^2}{\eta_{app}}, \tag{2}$$

$$Ne' = 70\,Re^{-1} + 25\,Re^{-0.6} + 1.5\,Re^{-0.2}, \tag{3}$$

where $\eta_{app}$ is the apparent dynamic viscosity of the culture. To consider the impact of the air−water interface on BUD-ELM assembly, the volumetric power input was multiplied by the volumetric mass transfer coefficient of oxygen, defined dimensionally by ref. 49, as:

$$k_L a = 0.5 \cdot d^{\frac{73}{36}} \cdot n \cdot d_0^{\frac{1}{4}} \cdot V_L^{-\frac{8}{9}} \cdot D^{\frac{1}{2}} \cdot v^{-\frac{13}{54}} \cdot g^{-\frac{7}{54}}, \tag{4}$$

where $k_L$ is the transfer coefficient of oxygen, $a$ is the oxygen transfer surface area, $d_O$ is the shaking orbit diameter, $D$ is the diffusion coefficient, $v$ is the kinematic viscosity, and $g$ is the acceleration of gravity. To unify size measurements across different flask sizes, a correction factor of $d^5$ was applied. This new parameter was dubbed the "modified volumetric power", defined as:

$$P_{v,a} = d^5 \cdot k_L a \cdot \left(\frac{P}{V_L}\right). \tag{5}$$

Calculations for $P_{v,a}$ assumed that the media parameters for viscosity $\eta_{app}$ and $v$ and the diffusion coefficient $D$ were equal to that of water at a standard growth temperature of 30 °C, which approximates culture conditions at inoculation. Constants were set to the following values:

$\eta_{app}$ = 7.97·10⁻⁴ Pa·s; $\rho$ = 995.67 kg/m³; $d_O$ = 0.05 m; $D$ = 2.5655·10⁻⁵ m²/s; $v$ = 8.005·10⁻⁷ m²/s; $g$ = 9.807 m/s².

**Parameter definitions and units.** $n$– shaking frequency (s$^{-1}$); $Ne'$ – modified Newton number (dimensionless); $P$ – power input (W); $Re$ – Reynold's number (dimensionless); $V_L$ – culture volume (m$^3$); $\eta_{app}$ – dynamic apparent viscosity (Pa•s); $\rho$ – liquid density (kg/m$^3$); $d_O$ – orbital shaking diameter (m); $D$ – diffusion coefficient (m$^2$/s); $\nu$ – kinematic viscosity (m$^2$/s); $g$ – acceleration of gravity (m/s$^2$).

## Rheological measurements
The rheological properties of BUD-ELMs produced from strains RCC002, RCC004 and MFm152 (original BUD-ELM, $\Delta ELP_{60}$, and $\Delta rsaA_{1-250}$ strains, respectively) were evaluated on a strain-controlled (ARIES G2) rheometer equipped with an 0.1 rad 8-mm diameter cone plate. BUD-ELMs were grown in standard conditions. An approximate volume of 100–200 μL of BUD-ELMs were collected into a 1.5 mL centrifuge tube and spun for 10 s at 3200 rcf with a mini centrifuge (VWR® – C0803). This allowed for the material to collect at the bottom as a homogeneous paste. The supernatant was removed and 150uL of fresh PYE were added on top of BUD-ELMs to prevent desiccation. Strain sweep experiments from 0.1 to 100% strain amplitudes were performed at a fixed frequency of 3.14 rad/s. Frequency sweep experiments from 100 to 0.1 rad/s were performed at a 0.35% strain amplitude. Data were acquired using TRIOS software (version 4.2.1.36612).

## Biosorption of Cd$^{2+}$ to BUD-ELMs
To measure the ability of BUD-ELMs to bind Cd$^{2+}$, the $\Delta SpyTag$ BUD-ELM strain was cultured in standard conditions. After growth for 24–48 h, BUD-ELMs were harvested into sterile 2 mL tubes and lyophilized for 5 h. Lyophilized BUD-ELMs were transferred to a metal-free 15 mL tube (VWR® Metal-Free Centrifuge Tubes, Polypropylene, Sterile) and incubated with 7 mL of 6 ppm CdCl$_2$ (Sigma Aldrich – 202908) in ddH2O for 90 min on an orbital shaker. After incubation, the Cd$^{2+}$ concentration of the supernatant was measured by ICP-MS. Specifically, 5 μL of the supernatant was diluted in 4.995 mL 1% HNO$_3$ with 5 μg/mL Indium (In), as standard for data analysis (Perkin Elmer N9303741). This diluted solution was run on a Perkin Elmer Nexion 300 ICP-MS with two isotopic measurements (Cd$^{2+}$ 111 and Cd$^{2+}$ 112) and In 115 as the internal standard. Data was acquired using Syngistix software.

## Functionalization of BUD-ELMs with PQQ-Glucose Dehydrogenase (GDH)
Single colonies of *E. coli* BL21(DE3) harboring pSMCAF032 and pSMCAF029, for expression of GDH and SpyCatcher-GDH, respectively, were inoculated in 25 mL of Terrific broth (TB) with 100 μg/mL ampicillin. After -16 h of growth at 37 °C and 250 rpm, cells were used to inoculate 0.5 L of TB with 0.02% antifoam (Antifoam 204) and 100 μg/mL ampicillin to a final OD$_{600}$ -0.05. The cultures were allowed to grow at 37 °C until the mid-log phase. Protein production was induced with 0.2% w/v L-arabinose with incubation at 30 °C for -17 h.

Cells were harvested by centrifugation at 8000 × $g$ for 5 min, resuspended in lysis buffer (0.01 M MOPS) and lysed using sonication. After centrifugation at 12,000 × $g$ for 1 h, the GDH (or SpyCatcher-GDH) in the supernatant was reconstituted by adding a final concentration of 3 mM Ca$^{2+}$ and 0.06 mM PQQ and incubated for 15 min at 4 °C. BUD-ELMs, washed once with 0.01 M PBS, were incubated with reconstituted and non-reconstituted cell lysates for 2 h at 4 °C and then washed three times with 0.01 M PBS. A small piece of functionalized BUD-ELMs (Supplementary Fig. 11) was used for the colorimetric assay to detect GDH activity.

## Colorimetric test to detect glucose dehydrogenase (GDH) activity
The activity of GDH functionalized material was quantified with a modified colorimetric 2,6-dichlorophenol (DCPIP) assay[50]. A reagent solution of 48 mL MOPS buffer (10 mM, pH 7.0, 47 mL), 1 mL DCPIP (20 mg dissolved in 5 mL of DI water), and 1 mL phenazine methosulfate (PMS) (45 mg dissolved in 5 mL of DI water) were prepared. Analytical samples (-3 uL) were mixed with the reagent to 190 μL. The reaction was initialized by adding 10 μL glucose (2 M). Glucose consumption was correlated to the consumption of DCPIP (2:1 ratio), which was quantified colorimetrically by absorption at 600 nm. A representative curve is shown in Supplementary Fig. 14. All the activity assays were performed at room temperature.

## Statistics and reproducibility
Figure 1c (left) is a representative example of over 100 independent experiments.

Figures 1e, 2c, d, and Supplementary Fig. 3 are representative examples of two independent experiments.

Figures 1c (right), d, 2a, and f, and Supplementary Figs. 1, 2, 4, 5, 7, 9, and 10 are representative examples of at least three independent experiments.

The blots shown in Fig. 2e and Supplementary Figs. 6 and 8 were repeated twice, each one with three independent samples.

## Reporting summary
Further information on research design is available in the Nature Research Reporting Summary linked to this article.

## Data availability
All the data generated in this study are provided in the Supplementary Information/Source Data file. Plasmids and strains can be provided upon request submitted to the corresponding author (Prof. Caroline Ajo-Franklin, cajo-franklin@rice.edu) for at least 10 years following the publication date. The transfer of the material will be initiated within two weeks from the first request. Source data are provided with this paper.

## Code availability
The codes used to quantify the flat surface area of the material are available to download on Githubs (https://github.com/CAJOlab/BUD-ELM-Image-Analysis).

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

## Acknowledgements

We thank Marimikel Charrier, Maria Orozco Hidalgo, and Dr. Vera Troselj for helpful conversations. This work was primarily supported by the Defense Advanced Research Projects Agency (Engineered Living Materials Program, C.M.A.-F.). Additional support was provided by the Cancer Prevention and Research Institute of

Texas (RR190063, C.M.A.-F.) and the Office of Naval Research (N00014-21-1-2362, C.M.A.-F.). Work at the Molecular Foundry was supported by the Office of Science, Office of Basic Energy Sciences, of the U.S. Department of Energy under Contract No. DE-AC02-05CH11231.

## Author contributions

Conceptualization: S.M., C.M.A.-F., R.F.T., D.L., K.R.R., and P.D.A. Methodology: S.M., R.F.T., R.C., S.S., J.S., D.L., and C.M.A.-F. Investigation: S.M., R.F.T., R.C., S.S., J.S., and D.L. Visualization: S.M., R.F.T., R.C., and D.L. Funding acquisition: C.M.A.-F., K.R.R., and P.D.A. Project administration: C.M.A.-F. Supervision: C.M.A.-F. and S.M. Writing—original draft: S.M., C.M.A.-F., and R.F.T. Writing—review and editing: S.M., C.M.A.F., R.F.T., R.C., S.S., J.S., D.L., K.R.R., and P.D.A.

## Competing interests

The authors declare no competing interests.
