## [Peer Review File · Nature Communications]

Reviewers' Comments:

Reviewer #1:

Remarks to the Author:

The paper presents a new class of ELM based on synthetic biopolymer matrices from *C. crescentus*. This is a useful addition to the field, especially in terms of expanding the toolbox of chassis strains, the types of engineered matrices that exist, and the contribution to understanding how matrix molecular features contribute to mechanical properties. The fact that the BUD-ELM generates spontaneously in the culture flask is also worthy of further development, and the authors are right to point out the lack of ELMs that can be fabricated without top-down processing and the addition of exogenous polymers. This makes the BUD-ELM system potentially useful as a launching point for ELMs that develop structure and function autonomously. There are some aspects of the paper that could be strengthened. The control over BUD-ELM morphology seems crude at the moment, and the value of the modeling seems unclear without a vision of how it would be used. The functional demonstrations of metal binding and enzymatic catalysis are de rigueur at this point in the ELM literature, and if they are to be used here the experiments should be explained better and comparisons to other methods would strengthen the paper. That said, I think this could be a valuable alternative to other methods for ELM biosynthesis.

- The statement "Engineering principles to achieve this are unknown, so most ELMs are microscopic and must be processed into macroscopic materials." is disputable because of the potentially broad definition of ELMs. It seems that several qualifiers would need to be used in this statement, including terms like "autonomously produced" and "genetically engineered". For example, several bacterial cellulose-based materials are generated autonomously on macroscopic scales.
- "However, these approaches have afforded little genetic control over the matrix composition and only ~20-30% changes in material mechanics". Our lab has reported larger increases in material mechanics through genetic control, for example 3-6x increases in hydrogel shear modulus through genetically encoded crosslinking (doi.org/10.1038/s41467-021-26791-x), and quite a bit of genetic control over matrix composition!
- Fig 1D and 1E have a typo – "strained"  "stained"
- Fig. 2B – what is the point of the DiO staining? Is there an expected lipid-based matrix component?
- Only secreted protein does not form the matrix. What is the nature of the matrix-displayed protein interaction? Would purify the protein to investigate self-assembly shed some light on this?
- The apparent encapsulation of the BUD-ELM in microbial cells (Fig 2A.B) is interesting. Is this due to the relative hydrophobicity of the ELP matrix? It would be interesting to know if the ELP-less BUD behaved similarly.
- Does shaking promote protein detachment from the cell surface through shear? Could sonication help control this process more precisely than the shaking speed of flasks?
- I have several questions about the mathematical modeling:
 - o What is its purpose? The conclusions that the authors reach about how shaking and flask diameter affect the size of BUD-ELMs seems like they could be supported decently by the phenomenological description of outcomes for different conditions. The model only seems useful if it could be predictive in some way to aid in design iteration, or if it shed some light on mechanisms that could not be readily observed phenomenologically. However, the paper does not show this.
 - o Perhaps the model could be better justified if the authors provide a forward-looking view of how it could be used to control BUD-ELM properties in more sophisticated ways.
- Fig. 4A – captions should have (top) and (bottom) rather than (left) and (right).
- For Cd²⁺ removal, it would be interesting to know how much of the removal is due to the BUD, as opposed to unmodified *C. crescentus* cells. It is unclear what is being reported in Figure 4F – what does the Flowthrough bar represent?
- What is the dry mass yield of BUD-ELMs per liter of culture?
- Although the foundational aspects of BUD-ELM development are recognized, it would be useful in the discussion section for the authors to give some indication of how the BUD-ELM might be uniquely useful for practical applications. For example, the storage modulus values the authors report for the delta-ELP BUDs seem significantly higher than others reported in the literature. Perhaps this could be leveraged for more mechanically robust functional materials of some kind. Perhaps there is some precedence in other ELM literature (bacterial cellulose?) for using

hydrodynamic forces to control morphology that could be used for BUDs with help from the modeling.

- Many of the morphological differences between BUD-ELM variants discussed in the paper do not come across so clearly in the figure images. For example, the materials in Figures 3A and 4A look quite similar to me.

Neel Joshi

Reviewer #2:

Remarks to the Author:

The fancy images of results look nice and the story and the flowing are also good. However, I was feeling their manuscript is pretty unkind for readers in overall, and overall quality is not enough to publish in Nature communications journal. Their big aim is good but it's lacked explanation in both of introduction and results section. Also, I think the main manuscript & supporting documents are still not formatted well for publication to this journal.

During reading the manuscripts, I was thinking that controls in overall experiments are not enough to demonstrate the results. I was also really wondering these all experiments are repeatable from individual cultures (biologically), and also other people can follow the experiments like that this group did.

Particular, I can't understand about Fig 2D image. There is no explaining how the antibody can bind BUD protein, and how we know it's actually BUD, even about the protein size. I couldn't find any information about it in the manuscript and even the method section.

They already have fancy results, but I hope this article will be published with more compelling storytelling and evidence.

REVIEWER COMMENTS

Reviewer #1 (Neel Joshi, Remarks to the Author):

The paper presents a new class of ELM based on synthetic biopolymer matrices from *C. crescentus*. This is a useful addition to the field, especially in terms of expanding the toolbox of chassis strains, the types of engineered matrices that exist, and the contribution to understanding how matrix molecular features contribute to mechanical properties. The fact that the BUD-ELM generates spontaneously in the culture flask is also worthy of further development, and the authors are right to point out the lack of ELMs that can be fabricated without top-down processing and the addition of exogenous polymers. This makes the BUD-ELM system potentially useful as a launching point for ELMs that develop structure and function autonomously. There are some aspects of the paper that could be strengthened. The control over BUD-ELM morphology seems crude at the moment, and the value of the modeling seems unclear without a vision of how it would be used. The functional demonstrations of metal binding and enzymatic catalysis are de rigueur at this point in the ELM literature, and if they are to be used here the experiments should be explained better and comparisons to other methods would strengthen the paper. That said, I think this could be a valuable alternative to other methods for ELM biosynthesis.

Response: We thank the Reviewer for his strong endorsement of our work and for pointing out areas in which the manuscript can be strengthened.

- The statement “Engineering principles to achieve this are unknown, so most ELMs are microscopic and must be processed into macroscopic materials.” is disputable because of the potentially broad definition of ELMs. It seems that several qualifiers would need to be used in this statement, including terms like “autonomously produced” and “genetically engineered”. For example, several bacterial cellulose-based materials are generated autonomously on macroscopic scales.

- “However, these approaches have afforded little genetic control over the matrix composition and only ~20-30% changes in material mechanics“. Our lab has reported larger increases in material mechanics through genetic control, for example 3-6x increases in hydrogel shear modulus through genetically encoded crosslinking (doi.org/10.1038/s41467-021-26791-x), and quite a bit of genetic control over matrix composition!

Response: We agree with the Reviewer that both of these statements need to more specifically define the type of ELM to be accurate and thank him for his insight here. We have revised the text to read:

Engineering principles to achieve this assembly are unknown^{11,12}. Therefore, most macroscopic ELMs have been produced by adopting a top-down approach (such as 3D printing) to incorporate living cells into a exogenous matrix^{6,13,14} or by processing microscopic ELMs that grow a synthetic biomolecular matrix into macroscopic materials¹⁵⁻¹⁹. The few autonomously produced, macroscopic ELMs have been created by genetically modifying existing nanocellulose matrices²⁰ or genetically manipulating mineralization of silica matrices²¹. However, these two approaches to

autonomously produced, macroscopic ELMs have afforded little genetic control over the matrix composition and only ~20-30% changes in material mechanics^{20,21}. This tunability is much more limited than the tunability of naturally-occurring materials, chemically synthesized materials, or macroscopic ELMs produced by processing^{22,23}.

• **Fig 1D and 1E have a typo – “strained”  “stained”**

Response: Thank you for these corrections. We have edited the Figure 1 legend accordingly.

• **Fig. 2B – what is the point of the DiO staining? Is there an expected lipid-based matrix component?**

Response: Thank you for this helpful question. We included the DiO staining in order to probe whether the matrix contained lysed cells. If it did, we expected to see DiO staining of cell membranes co-localize with the matrix staining. However, if the matrix did not have a significant number of lysed cells, we expected little co-localization of the DiO staining with the matrix staining.

We have added the following text to the manuscript to explain the rationale for the DiO staining: “The absence of DiO staining excludes the hypothesis that the BUD-ELM matrix contains remains of lysed cells.”

• **Only secreted protein does not form the matrix.**

Response: We thank the Reviewer for these helpful questions, as they highlight important points where our manuscript is unclear. The Reviewer is incorrect in asserting that ‘*only secreted protein does not form the matrix.*’ The $\Delta rsaA_{1-250}$ BUD-ELM strain secretes the BUD protein, but does not display it on the cell surface. The $\Delta rsaA_{1-250}$ BUD-ELM also forms macroscopic BUD-ELMs, however, the resulting material has a much lower cell content than the original BUD-ELM strain (revised Fig 2f). These data show that the secreted BUD protein self-interacts to form a soft solid material and that the interaction between displayed BUD proteins and the matrix is necessary for cell-rich materials to be formed. To clarify these very important points, we have introduced Figure 2f to compare the *rsaA* locus, presence of material, and microscopic morphology of the BUD-ELM, ΔELP_{60} BUD-ELM, and $\Delta rsaA_{1-250}$ BUD-ELM strains. We have also revised the results to include the following text:

“Having established the roles of the BUD protein in BUD-ELM assembly, we next sought to understand the role of each domain of the BUD protein in assembly. To do so, we generated an additional strain lacking the ELP_{60} (ΔELP_{60} BUD-ELM strain, Fig. 2f – left panel, top image) and compared it to the original BUD-ELM strain (Fig. 1b). The RsaA C-terminal domain could not be deleted because it is known to be essential for extracellular secretion. We observed that the ΔELP_{60} BUD-ELM strain (Fig. 2f – left panel, bottom-left image) forms BUD-ELMs that are very similar to the original BUD-ELM in morphology and color; optical microscopy also confirms that both are cell-rich materials (Fig. 2f – left panel, bottom images). Moreover, confocal microscopy shows that - despite the removal of the central ELP_{60} domain - single cells are still surrounded by a layer of BUD protein (Fig. S7). While the SpyCatcher-GFP staining is less intense (laser intensity was increased by 25% to visualize it), whole-cell immunoblotting indicates the amount of BUD protein attached to ΔELP_{60} cells is at least comparable to, if not greater, than the original BUD-ELM strain (Fig. S8). This indicates the BUD protein lacking the ELP_{60} region is less solution-accessible than the original BUD protein. Taken together, these data show that the ELP_{60} domain promotes solution accessibility of BUD protein.

We also examined the role of the anchoring domain in material formation by using a previously described strain that lacks RsaA₁₋₂₅₀, but contains ELP₆₀ and RsaA₆₉₀₋₁₀₂₆²⁹ (Fig. 2f – right panel, top image). This Δ rsaA₁₋₂₅₀ strain, where the BUD protein is only secreted but not displayed, formed centimeter-scale materials (Fig. 2f – right panel, bottom-left image). These materials are much lighter in color than the original strain, suggesting they contain fewer cells. Indeed, optical microscopy (Fig. 2f – right panel, bottom-right image) showed that this Δ rsaA₁₋₂₅₀ BUD-ELM has many fewer cells than the original BUD-ELM. Together, these data indicate that secreted BUD proteins critically enable centimeter-scale BUD-ELM assembly. Surface-displayed BUD proteins, on the other hand, allow formation of cell-rich BUD-ELMs by facilitating cell-cell and cell-matrix interactions. Since we observe large cell-cell aggregates in the original BUD-ELM strain (Fig. 1c – left) that are spatially distinct from the secreted matrix (Fig. 2a), we suggest that this aggregation is promoted by high-density protein display. Thus, this work provides genetic design rules for both cell-rich and matrix-rich macroscopic engineered living materials that autonomously form.”

•What is the nature of the matrix-displayed protein interaction? Would purify[ing] (sic) the protein to investigate self-assembly shed some light on this?

Response: The nature of self-interactions within the BUD protein has not been entirely elucidated. Prior reports demonstrate that RsaA₆₉₀₋₁₀₂₆ self-aggregates³⁹, but the nature of the interaction has not been characterized. Moreover, ELPs are well-known to self-interact in a temperature and sequence-dependent manner⁴⁴. While biophysical studies would likely elucidate the nature of these interactions, preliminary investigations in our lab have shown that purification and *in vitro* characterization of the BUD protein is quite challenging. While characterization of the self-interactions in the BUD protein is outside of the scope of this initial study, future studies will use *in vitro* and/or *in vivo* experiments to determine the nature and strength of these self-interactions.

We have revised the discussion to include these important points, as follows: “Another design rule that will be critical to understand and explore in future work is the nature and strength of the self-interactions in the BUD protein. We selected the RsaA₆₉₀₋₁₀₂₆ and ELP₆₀ domains because prior reports demonstrate they can self-aggregate^{39,44}. However, additional studies are needed to identify the nature of self-interactions and their strengths in the existing BUD protein and the range of self-interactions that permit assembly of macroscopic materials.”

• The apparent encapsulation of the BUD-ELM in microbial cells (Fig 2A.B) is interesting. Is this due to the relative hydrophobicity of the ELP matrix? It would be interesting to know if the ELP-less BUD behaved similarly.

Response: We agree with the Reviewer that this self-encapsulation is an interesting aspect of our system. To address whether the ELP-less BUD ELM encapsulated cells, we performed new experiments using confocal microscopy and whole cell immunoblotting. Confocal microscopy of the Δ ELP₆₀ BUD-ELMs stained with SpyCatcher-GFP (**new Figure S7**) confirms the cells are encapsulated by the ELP-less BUD protein. Because the stained GFP intensity in the Δ ELP₆₀ BUD-ELM was significantly decreased relative to the original BUD-ELM strain, we also used whole cell immunoblotting to qualitatively compare the amount of surface-attached BUD protein. The Δ ELP₆₀ BUD-ELM strain displays more BUD protein than the BUD-ELM strain (**new SI Figure**

S8). These data show that the ELP-less BUD ELM cells are fully encapsulated by the BUD-protein. These data also strongly suggest that hydrophobicity from ELP₆₀ does not impact encapsulation, but it does improve GFP-staining by increasing the accessibility of SpyTag.

We have also revised the description of these results as follows:

“Moreover, confocal microscopy shows that - despite the removal of the central ELP₆₀ domain - single cells are still surrounded by a layer of BUD protein (Fig. S7). While the SpyCatcher-GFP staining is less intense (we had to increase the laser intensity by 25% to visualize it), whole-cell immunoblotting indicates the amount of BUD protein attached to the Δ ELP₆₀ cells is at least comparable to, if not greater, than the original BUD-ELM strain (Fig. S8). This indicates the BUD protein lacking the ELP₆₀ region is less solution-accessible than the original BUD protein. Taken together, these data show that the ELP₆₀ domain promotes solution accessibility of BUD protein.”

• **Does shaking promote protein detachment from the cell surface through shear? Could sonication help control this process more precisely than the shaking speed of flasks?**

Response: The possibility that shaking affects the protein detachment from the cell surface is a very interesting hypothesis. Currently, we have evidence that the BUD protein is released into the culture media under both shaking and static conditions (Fig. 2D, **now Fig. 2e**), indicating that shaking is not required for protein detachment. However, several technical challenges make it difficult to quantify the effect of shaking on protein detachment. In particular, shaking affects cell growth - which might increase the amount of protein synthesized - and material formation - which can sequester detached protein. Quantitatively untangling these potentially interfering effects is beyond the scope of this paper.

Sonication could be a very effective method to release the BUD protein from the cell surface. However, we have found that it also greatly affects cell viability, which impairs our ability to make materials that still contain living cells. Additionally, sonication would also be an additional treatment/processing step, which our work seeks to avoid.

We have revised the text to highlight that BUD protein detachment is independent of shaking, as follows:

“Hypothesizing that BUD proteins in this layer might be released from the cell surface as the layer thickens, we checked the extracellular medium of BUD-ELM cultures for the presence of the BUD protein by immunoblotting against the FLAG tag. We observed a band corresponding to the BUD protein at ~102 kDa, indicating that it is indeed present in the extracellular media. Note that wild-type RsaA migrates at a higher than expected apparent molecular weight (observed 113 kDa vs. expected 98 kDa³⁰), and the BUD protein shows this same apparent difference in molecular weight (observed 102 kDa vs. expected 86 kDa). Interestingly, we found the same amount of secreted BUD protein in both shaken and static cultures (Fig. 2e), indicating that release of the BUD protein into the medium is independent of shaking. Nonetheless, these results demonstrate that the BUD protein is simultaneously present as a surface-displayed matrix protein and a secreted matrix protein during material formation.”

• **I have several questions about the mathematical modeling:**

o **What is its purpose? The conclusions that the authors reach about how shaking and flask diameter affect the size of BUD-ELMs seems like they could be supported decently by the phenomenological description of outcomes for different conditions. The model only seems useful if it could be predictive in some way to aid in design iteration, or if it shed some light on mechanisms that could not be readily observed phenomenologically. However, the paper does not show this.**

o **Perhaps the model could be better justified if the authors provide a forward-looking view of how it could be used to control BUD-ELM properties in more sophisticated ways.**

Response: We thank the Reviewer for these excellent points and very well-posed questions.

We now clarify that the primary goal of the modeling is to enable a phenomenological prediction of growth conditions that yield the largest pieces of material. New data (**revised Figure 3d**) show that the model accurately predicts the growth conditions to maximize the apparent material size for cultures grown in 500 mL flasks. These new data demonstrate the predictive nature of this model, justifying its development and presentation.

We have revised the text to more clearly justify the model and to demonstrate that the model accurately predicts the growth conditions to maximize the material size, as follows:

“To test whether this phenomenological model could accurately predict the size of material grown in larger flasks, we calculated shaking conditions for 500 mL flasks that would match $P_{V,A}$ values inside and outside the optimal range and used these conditions to grow BUD-ELM cultures in 500 mL flasks (Table S4). In agreement with our model, cultures grown in the optimal $P_{V,A}$ range yielded material with large apparent sizes, while cultures outside this range yielded smaller material (Fig. 3d). This evidence indicates that the optimal $P_{V,A}$ range can be applied to large-sized material production across different culture sizes and demonstrates our model can be used to scale-up cultures for BUD-ELM production.”

• **Fig. 4A – captions should have (top) and (bottom) rather than (left) and (right).**

Response: Thank you. These errors have been corrected.

• **For Cd²⁺ removal, it would be interesting to know how much of the removal is due to the BUD, as opposed to unmodified *C. crescentus* cells. It is unclear what is being reported in Figure 4F – what does the Flowthrough bar represent?**

Response: We thank the Reviewer for these insightful questions.

In the Cd²⁺ removal experiments, a solution of Cd²⁺ was incubated with and without the BUD-ELM. The flowthrough bar represents the Cd²⁺ concentration of the solution without BUD-ELM. We recognize this terminology is confusing and have revised the Figure and text to label this as ‘without BUD-ELM.’

To compare the removal of Cd²⁺ by wild-type cells and BUD-ELM cells, we would need to determine the number of cells in the BUD-ELM materials. This is technically very challenging because, at present, all methods we have identified to dissolve the material also lyse the cells. However, our intent is not to demonstrate that our material outperforms single cells. (Many bacteria absorb heavy metals due to non-specific interactions with chemical groups on the cell surface⁴². Rather, we argue that a macroscopic, cadmium-absorbing solid material is a much more useful tool for solution decontamination than a suspension of single cells.

Our claim was not well explained in the original manuscript and therefore we have rephrased it for clarity, as follows:

“Since many forms of bacterial biomass non-specifically absorb heavy metals⁴², we hypothesized that the BUD-ELM could remove Cd²⁺ from solution. When 0.013±0.007g of Δ*SpyTag* BUD-ELM was incubated for 90 min with a CdCl₂ solution of 6 ppb–1 ppb above the Environmental Protection Agency (EPA) limit–90 ±5 % of cadmium was removed (Fig. 4e). While this material is not designed to have a larger sorption capacity than wild-type *C. crescentus* cells, these data show that the BUD-ELM has potential to be a much more useful tool for heavy metal removal than a suspension of single cells by virtue of being a macroscopic, solid material.”

• **What is the dry mass yield of BUD-ELMs per liter of culture?**

Response: We agree with the Reviewer that this is a critical parameter for the characterization of our living. In a set of new experiments, we grew, collected, lyophilized, and weighed BUD-ELMs from eight biological replicates. The average dry mass was 350 ± 302 mg dry mass/L culture. While the mass yield varies significantly between replicates, the average mass yield of the BUD-ELMs is ~6x greater than the mass yield of secreted protein from *C. crescentus*²⁹. We added this new data in the main text and included the experimental description in the methods section.

• **Although the foundational aspects of BUD-ELM development are recognized, it would be useful in the discussion section for the authors to give some indication of how the BUD-ELM might be uniquely useful for practical applications. For example, the storage modulus values the authors report for the delta-ELP BUDs seem significantly higher than others reported in the literature. Perhaps this could be leveraged for more mechanically robust functional materials of some kind. Perhaps there is some precedence in other ELM literature (bacterial cellulose?) for using hydrodynamic forces to control morphology that could be used for BUDs with help from the modeling.**

Response: We thank the Reviewer for both highlighting the need for foundational ELM development and the need to suggest how this material might be uniquely useful. We now compare the mechanical properties of the BUD-ELMs to other ELMs with synthetic biomolecular matrices. We also point out that the incorporation of elastin-like polypeptides may allow this material to be developed into an elastomer. We have added new text to the discussion to describe these features:

The BUD-ELM variants described herein have a storage moduli that ranges between 13 kPa, comparable to nanocellulose-based materials, and 0.5 kPa, comparable to printed curli fiber-based materials. Introducing sites for chemical crosslinking into the ELP domain could allow the BUD-ELMs to be developed into elastomers⁴⁵.

• **Many of the morphological differences between BUD-ELM variants discussed in the paper do not come across so clearly in the figure images. For example, the materials in Figures 3A and 4A look quite similar to me.**

Response: We agree with the Reviewer that the morphological differences between the materials are not clear from the provided images. To address this point, we have revised Figure 2 to provide a side-by-side comparison of the cm-scale morphology and micron-scale morphology in Figure 2f. We have also revised the main text to highlight the key features which differ between the BUD-ELM variants as indicated in the prior comment.

Reviewer #2 (Remarks to the Author):

The fancy images of results look nice and the story and the flowing are also good. However, I was feeling their manuscript is pretty unkind for readers in overall, and overall quality is not enough to publish in Nature communications journal. Their big aim is good but it's lacked explanation in both of introduction and results section.

Response: We appreciate the Reviewer's support for the technical quality of our data and presentation of our results. We agree with the Reviewer that a more extensive introduction and discussion is needed to make the manuscript more accessible to the broad audience of *Nature Communications*. Accordingly, we have significantly expanded the introduction to include a description of macroscopic materials formed by processing and a brief summary of our work:

“Engineering principles to achieve this assembly are unknown^{11,12}. Therefore, most macroscopic ELMs have been produced by adopting a top-down approach (such as 3D printing) to incorporate living cells into a exogenous matrix^{6,13,14} or by processing microscopic ELMs that grow a synthetic biomolecular matrix into macroscopic materials¹⁵⁻¹⁹. The few autonomously produced, macroscopic ELMs have been created by genetically modifying existing nanocellulose matrices²⁰ or genetically manipulating mineralization of silica matrices²¹. However, these two approaches to autonomously produced, macroscopic ELMs have afforded little genetic control over the matrix composition and only ~20-30% changes in material mechanics^{20,21}. This tunability is much more limited than the tunability of naturally-occurring materials, chemically synthesized materials, or macroscopic ELMs produced by processing^{22,23}.”

We posit that new strategies for developing synthetic biomolecular matrices to self-assemble bacteria into macroscopic ELMs can be informed by prior work on surface-engineered bacteria and surface-modified colloidal particles. The surface of *Escherichia coli* has been engineered to display interacting proteins, such as leucine zippers²⁴ or antigen-nanobody pairs²⁵, via outer membrane proteins. Engineered strains that display interacting pairs will self-assemble into cell-cell aggregates that flocculate^{24,25}, however, these aggregates are microscopic and must be processed to form larger materials¹⁸. In contrast, micron-sized colloidal particles (typically polystyrene) that display DNA have been programmed to self-assemble into both microscopic²¹ and macroscopic crystalline solids²⁶. Over two decades of work on these systems has established central principles that underlie their self-assembly²⁷. One of these central principles is that the interactions between particles must be high density, e.g. 1 DNA per 27²⁶. Since the outer membrane proteins used for bacterial adhesins are displayed at ~5% of this density, i.e. 1 nanobody

per 640 nm²²⁸, we hypothesized that a matrix composed of self-interacting proteins displayed on bacteria at high density could lead to formation of macroscopic solid materials.

We have previously engineered the surface layer (S-layer) of the oligotrophic bacterium *Caulobacter crescentus* for high-density protein display²⁹ and biopolymer secretion²³. The S-layer forms a 2D crystalline layer on the extracellular surface of *C. crescentus*, opening the possibility of displaying proteins at a density of up to 1 protein per 70 nm²³⁰. Leveraging this prior work, here we describe the autonomous formation of a macroscopic living material from *C. crescentus* engineered to display a synthetic, self-interacting, protein matrix based on the S-layer scaffold. We demonstrate that the stiffness of this material can be genetically controlled over a factor of ~16x. We also describe unexpected findings indicating that the protein matrix plays a multifaceted role in material formation and that material assembly occurs through a multi-step process mediated by the air-water interface.”

We also expanded our explanation of the results in several locations suggested by Reviewer #1 (see prior comments).

Also, I think the main manuscript & supporting documents are still not formatted well for publication to this journal.

Response: The manuscript was originally formatted for *Nature* and transferred to *Nature Communications*. With the opportunity to submit a revised manuscript to *Nature Communications*, we have added a more thorough explanation of our work and re-formatted the manuscript per *Nature Communications* guidance.

During reading the manuscripts, I was thinking that controls in overall experiments are not enough to demonstrate the results.

Response: We respectfully disagree with the Reviewer on this very important point. Our key conclusions are three-fold, that: i) expression of the BUD protein enables macroscopic material formation through a proteinaceous matrix, ii) formation of the BUD-ELMs occurs through a multi-step process that depends on physical parameters, and iii) that the mechanical and catalytic properties of BUD-ELMs can be tuned genetically. These conclusions are supported by a variety of control experiments. For the first conclusion, we demonstrate that the negative control - the parental strain of the BUD-ELM strain - does not form macroscopic material under identical growth conditions (Fig S1). We also use staining with only GFP as a negative control to demonstrate that the staining of the matrix is specific, i.e. is mediated by the SpyTag-SpyCatcher interaction (Fig. S2). For the second conclusion, cultures that contain a surfactant are used as a control to demonstrate that formation of a pellicle is necessary to form material (Fig. S9). Also, cultures that are not shaken are used as controls to support the conclusion that shaking is necessary for BUD-ELM formation (Fig. S10). For the third conclusion, we use the GDH lysate as a negative control to differentiate between catalytic activity encoded by the presence of SpyTag (Fig. 4f). We also provide additional control experiments showing cell lysates containing SpyCatcher-GDH and

GDH have equivalent activity (Fig. S14). These are just a few highlights of the many controls we used to support our conclusions.

To address this point, we have added a new paragraph in the discussion that summarizes our conclusions and key experimental results supporting those conclusions, as follows:

“In summary, we developed macroscopic living materials that autonomously grow from engineered bacteria and that can be genetically-encoded to have a wide range of mechanical properties. Specifically, we show that expression of a self-interacting protein - the BUD protein - enables macroscopic material formation (Fig. 1, Fig. S1). When displayed on the cell surface, the BUD protein mediates drives cell-cell aggregation; when secreted into the media, the BUD protein forms an extracellular matrix that binds these aggregates into a centimeter-scale structure (Fig. 2). Assembly of these ELMs starts with growth of the engineered strain as a predominately planktonic culture, followed by formation of a pellicle and its ultimately collapse into a final material (Fig 3). Importantly, understanding of these design and assembly rules enabled us to alter the stiffness of these ELMs by ~16-fold and to imbue them with catalytic properties (Fig. 4).”

I was also really wondering these all experiments are repeatable from individual cultures (biologically), and also other people can follow the experiments like that this group did.

Response: We strongly agree with the Reviewer that repeatability is a key concern in the field of engineered living materials. In our view, one of the greatest strengths of our work is the highly reproducible, autonomous formation of BUD-ELMs. The data in Figures S11b and 3d alone characterizes material formation from 107 biological replicates. Our work shows that growing BUD-ELMs from an engineered strain of *C. crescentus* requires only control of the temperature, media composition, flask and culture volume, shaking speed, and shaking orbit. Beyond these parameters, the only other requirement was that the strains be handled per standard microbiological practice. Under these conditions, all of our attempts to form BUD-ELMs were successful, including experiments performed by different authors at different institutions (D. Li at Lawrence Berkeley National Laboratory; S. Molinari, R.T. Tesoriero, and S. Sridhar at Rice University). Additional details on the reproducibility of our experiments are provided in the transparent reporting form.

To address this important point, we have added the following text to the discussion:

One of the key advantages of the *C. crescentus* BUD-ELM platform developed herein is the highly reproducible, autonomous formation of engineered living materials. Growing BUD-ELMs from an engineered strain of *C. crescentus* requires only control of the temperature, media composition, flask and culture volume, shaking speed, and shaking orbit. We envision this simplicity will enable ready adoption of this platform by other researchers.

Particular, I can't understand about Fig 2D image. There is no explaining how the antibody can bind BUD protein, and how we know it's actually BUD, even about the protein size. I couldn't find any information about it in the manuscript and even the method section.

Response: We thank the Reviewer for identifying this point of confusion. The immunoblot in Fig. 2d (now Fig. 2e) was probed with a monoclonal ANTI-FLAG® M2-Peroxidase antibody. This

information was previously only listed in the supplementary information; we have revised the manuscript to specify this information both in the main text and in the legend of Fig. 2.

They already have fancy results, but I hope this article will be published with more compelling storytelling and evidence.

Response: We appreciate the critical feedback that the manuscript would be improved with more elaboration in the introduction and results. As noted in the prior comments, we have extensively lengthened the manuscript to provide this additional information.

Reviewers' Comments:

Reviewer #1:

Remarks to the Author:

The revised manuscript addresses all of my previous critiques and I think the updates strengthen it considerably. A few other comments are below. The authors can use them as they see fit, but I don't think they need to hold up the publication of the manuscript, nor do they need any further review from me. Congrats to the authors on this nice work!

- A cartoon-like diagram of the *C. crescentus* cell surface with the BUD protein attached might be helpful for a broad audience to understand the connectivity of the ELM system. However, I recognize that such a diagram will necessarily be a bit speculative without more structural information.
- Line 56: "interactions between particles must be high density". This is an awkward sentence construction. Perhaps instead: "interactions between particle must be mediated by high-density surface modifications"
- Fig S1 is strange – it is just a monochrome square with a scale bar. It needs further context – what are we looking at?
- Fig S12 seems like it is labeled incorrectly in the caption as a "stress-strain curve". It should say "oscillatory strain sweep" or something similar.

Neel Joshi

Reviewer #2:

Remarks to the Author:

Their edited manuscript seems much kindness with more explanations than previous version. Particular, I was wondering how readers having no information about the antibody can know the Anti-Flag antibody can bind to only BUD proteins, not other proteins in the strain, but now it changed be much clear with the Fig S6 figure and provided protein sizes.

However, I still would like to suggest that they consider below things.

In the introduction section, you mentioned "However, these two approaches to autonomously produced, macroscopic ELMs have afforded little genetic control over the matrix composition and only ~20-30% changes in material mechanics"

However, I don't agree with this sentence. In the research of reference 21 (Kang et al, 2021), we have used a self-assembly scaffold protein and a biomineralization tag to allow a cross-link with silica, and the study is oriented toward fabricating genetically controllable biomaterial.

I agree that our material properties increased only ~1.4-fold in storage moduli and it's no significant changes compared to the control (ref 21). However, I would like to point out this sentence. The author demonstrated mechanical changes ~3.4 fold but calculated as ~300% unlike those presented as 20~30% for the reference. I strongly suggest that they change this sentence more in the introduction section so that there is no controversy.

Additionally, I would like to ask how the author calculated "16-fold" for mechanical properties. What did you use value for this calculation? Could you add this information in the result section (page 12)?

Two minor points about the changes are, could you make higher resolution image for Fig S10? It seems changed to a lower quality image during editing. Second is, for better figure of Fig S11, how about making to stand in the same left line for the graphs a (left) and b, and applying same font size of graph (each y-axis, x-axis)?

REVIEWERS' COMMENTS

Reviewer #1 (Remarks to the Author):

The revised manuscript addresses all of my previous critiques and I think the updates strengthen it considerably. A few other comments are below. The authors can use them as they see fit, but I don't think they need to hold up the publication of the manuscript, nor do they need any further review from me. Congrats to the authors on this nice work!

We thank the reviewer for such a strong endorsement of our work, and for all the insightful comments that contributed to strengthening the quality and clarity of this manuscript.

- *A cartoon-like diagram of the *C. crescentus* cell surface with the BUD protein attached might be helpful for a broad audience to understand the connectivity of the ELM system. However, I recognize that such a diagram will necessarily be a bit speculative without more structural information.*

We thank the reviewer for this insightful suggestion. We agree that a cartoon depicting the redesigned cell surface of *C. crescentus* would make the manuscript more accessible to a broad audience. We, therefore, decided to include one and revised Fig.1 accordingly.

- *Line 56: “interactions between particles must be high density”. This is an awkward sentence construction. Perhaps instead: “interactions between particle must be mediated by high-density surface modifications”*

We agree with the reviewer that this sentence may be confusing and revised as suggested.

- *Fig S1 is strange – it is just a monochrome square with a scale bar. It needs further context – what are we looking at?*

We agree with the reviewer that Fig. S1 does not clearly represent what was intended. For this reason, we provided a better picture and updated the figure caption, as follows:

Fig. S1. The wild-type strain does not form macroscopic aggregates. Representative image of wild-type *C. crescentus* strain (Mfm126) grown under standard conditions. The image was taken from the bottom of a 250 mL flask and shows the typical turbidity of a saturated culture of *C. crescentus* with no visible cellular aggregates. The scale bar is 1 cm.

- *Fig S12 seems like it is labeled incorrectly in the caption as a “stress-strain curve”. It should say “oscillatory strain sweep” or something similar.*

We thank the reviewer for catching the incorrect labeling of Fig. S12. We revised the figure caption as follows:

Fig. S12. Oscillatory strain sweep. Strain sweep measurements were acquired from 0.1% to 100% strain amplitude at a constant frequency of 3.14 rad/s. Error bars are centered on the mean value and represent 95% confidence intervals of at least five samples. From the amplitude sweep curves, we identified the linear viscoelastic region of the three

BUD-ELMs and set the strain used to collect frequency sweep data (Fig. S13) to 0.35%. Source data are provided as a Source Data file.

Neel Joshi

Reviewer #2 (Remarks to the Author):

Their edited manuscript seems much kindness with more explanations than previous version. Particular, I was wondering how readers having no information about the antibody can know the Anti-Flag antibody can bind to only BUD proteins, not other proteins in the strain, but now it changed be much clear with the Fig S6 figure and provided protein sizes.

We thank the reviewer for their strong endorsement of the revised manuscript and for insightful comments.

However, I still would like to suggest that they consider below things.

- In the introduction section, you mentioned "However, these two approaches to autonomously produced, macroscopic ELMs have afforded little genetic control over the matrix composition and only ~20-30% changes in material mechanics" However, I don't agree with this sentence. In the research of reference 21 (Kang et al, 2021), we have used a self-assembly scaffold protein and a biomineralization tag to allow a cross-link with silica, and the study is oriented toward fabricating genetically controllable biomaterial. I agree that our material properties increased only ~1.4-fold in storage moduli and it's no significant changes compared to the control (ref 21). However, I would like to point out this sentence. The author demonstrated mechanical changes ~3.4 fold but calculated as ~300% unlike those presented as 20~30% for the reference. I strongly suggest that they change this sentence more in the introduction section so that there is no controversy.*

We thank the reviewer for identifying this clarification. We have revised this sentence in the introduction to read:

“However, these two approaches to autonomously produced, macroscopic ELMs have afforded little genetic control over the mechanical properties, e.g. **~1.2-1.4 fold change in the storage modulus.**”

- Additionally, I would like to ask how the author calculated "16-fold" for mechanical properties. What did you use value for this calculation? Could you add this information in the result section (page 12)?*

We thank the reviewer for identifying this statement as a potential point of confusion.

To determine the maximum fold-change in storage modulus between our genetic variants, we divided the storage modulus of the ΔELP_{60} BUD-ELMs by the storage modulus of the $\Delta rsaA_{1-250}$ BUD-ELMs. Similarly, we calculated the maximum fold-change in loss modulus

by dividing the loss modulus of the ΔELP_{60} BUD-ELMs by the loss modulus of the $\Delta rsaA_{1-250}$ BUD-ELMs.

We have revised the results section to include this information as follows:

"For a central value of angular frequency of 10 rad/s (Fig. 4a), the storage modulus of ΔELP_{60} BUD-ELMs is increased by **4.4-fold** of the original BUD-ELM, whereas the loss modulus increased by **4.0-fold**. Conversely, the $\Delta rsaA_{1-250}$ BUD-ELMs show a **3.2-fold** and **6.3-fold** lower G' and G'' , respectively, relative the original BUD-ELM. Comparing the ΔELP_{60} BUD-ELMs to the $\Delta rsaA_{1-250}$ BUD-ELMs, we observe that these genetic changes can modulate the storage modulus and the loss modulus over **14-fold** and **25-fold**, respectively. We speculate that the increased stiffness of the ΔELP_{60} BUD-ELMs reflects the removal of a long elastic linker, the ELP_{60} , from the BUD protein forming this cellular material. On the other end, we suggest $\Delta rsaA_{1-250}$ BUD-ELMs are less stiff due to the lack of crosslinking among cells and between the matrix and the cells. Overall, these results demonstrate that we can control BUD-ELMs mechanical properties over a **25-fold** range through genetic modification of the matrix-forming BUD-protein. "

Two minor points about the changes are, could you make higher resolution image for Fig S10? It seems changed to a lower quality image during editing.

We thank the reviewer for noticing that the figure was accidentally attached to the document with a low resolution. We replaced it with a higher-resolution version of the same image.

Second is, for better figure of Fig S11, how about making to stand in the same left line for the graphs a (left) and b, and applying same font size of graph (each y-axis, x-axis)?

We thank the reviewer for his comments on figure S11. We agree that the graphs should have the same font size and we updated the figure accordingly. However, we decided to keep panel **a** and **b** in their original position to allow for a bigger size of the graphs, that in our opinion improves the clarity of the figure.